# Opposing JAK-STAT and Wnt signaling gradients define a stem cell domain by regulating differentiation at two borders

David Melamed, Daniel Kalderon*

Department of Biological Sciences, Columbia University, New York, United States

**Abstract** Many adult stem cell communities are maintained by population asymmetry, where stochastic behaviors of multiple individual cells collectively result in a balance between stem cell division and differentiation. We investigated how this is achieved for *Drosophila* Follicle Stem Cells (FSCs) by spatially-restricted niche signals. FSCs produce transit-amplifying Follicle Cells (FCs) from their posterior face and quiescent Escort Cells (ECs) to their anterior. We show that JAK-STAT pathway activity, which declines from posterior to anterior, dictates the pattern of divisions over the FSC domain, promotes more posterior FSC locations and conversion to FCs, while opposing EC production. Wnt pathway activity declines from the anterior, promotes anterior FSC locations and EC production, and opposes FC production. The pathways combine to define a stem cell domain through concerted effects on FSC differentiation to ECs and FCs at either end of opposing signaling gradients, and impose a pattern of proliferation that matches derivative production.

*For correspondence:
ddk1@columbia.edu

## Introduction

The physiological role of each type of adult stem cell is to maintain appropriate production of a restricted set of cell types throughout life (*Clevers and Watt, 2018*; *Post and Clevers, 2019*). To accomplish this objective, a sufficient population of stem cells must itself be maintained. Consequently, there must be some mechanism that balances stem cell proliferation and differentiation. The balance need not be precise or without fluctuations, especially if the stem cell population is large and therefore not at risk of temporary insufficiency or extinction. However, if the number of stem cells is held roughly constant over time, then an unchanging anatomy can provide a constant environment for regulating and matching stem cell divisions and differentiation.

The balance between stem cell division and differentiation can operate at the single-cell level or at the community level (*Jones, 2010*; *Mesa et al., 2018*; *Snippert et al., 2010*). If each stem cell repeatedly divides to produce a stem cell and a differentiated product ('invariant single-cell asymmetry'), the rate of division must simply be matched to the required supply of product cells. More commonly, however, a group of stem cells in a given location is maintained by 'population asymmetry', where individual stem cells exhibit non-uniform, stochastic behaviors and differentiation is commonly not temporally or mechanistically linked to division of the same stem cell (*Reilein et al., 2018*; *Ritsma et al., 2014*; *Rompolas et al., 2016*; *Simons and Clevers, 2011*). The behavior of such stem cells is likely guided substantially by extracellular signals and it is commonly presumed that regulation is achieved substantially by defining a compartment with fixed dimensions that can maintain stem cells. However, very little is known about how extracellular signals define niche space and the number of stem cells accommodated, how they affect stem cell division and differentiation, and whether they co-ordinate those two fundamental behaviors. *Drosophila* ovarian Follicle Stem Cells (FSCs) provide an outstanding paradigm to pursue these questions.

FSCs were first defined as the source cells for the Follicle Cell (FC) epithelium that surrounds each egg chamber (*Margolis and Spradling, 1995*). An egg chamber buds from the germarium of each

**eLife digest** Adult organisms contain a variety of cells that are routinely replaced using adult stem cells which can generate the cells of a specific tissue. These stem cells are often clustered into small groups, where combinations of chemical signals from nearby cells can encourage each stem cell to divide or 'differentiate' into another type of cell. These different signals must somehow balance stem cell division and differentiation to maintain the size and shape of the community.

The ovary of an adult fruit fly contains a group of adult stem cells called follicle stem cells, or FSCs for short. FSCs support the continual production of eggs by supplying two types of cell from opposite faces of the stem cell cluster: dividing follicle cells emerge from the back of the cluster and guide late egg development, while non-dividing escort cells come from the front and guide early egg development. Two of the signals that control FSCs are graded over the cluster. JAK-STAT signaling is strongest in the follicle cell territory and gradually declines towards the front, while Wnt signaling is strongest in escort cells and absent from early follicle cells. However, it was unclear how the gradients of these two signals maintain the FSC population and control the formation of follicle and escort cells.

To answer this question, Melamed and Kalderon used genetic engineering to modify the strength of these two signals. The experiments measured how this affected the rate at which FSCs divide and are converted into follicle or escort cells. Melamed and Kalderon found that the strength of JAK-STAT signaling dictated division rates, which may explain why the rate cells divide varies across the FSC cluster and escort cells do not divide at all. JAK-STAT signaling also stimulated FSCs to become follicle cells and opposed their conversion to escort cells. Conversely, stronger Wnt signaling favored the production of escort cells and inhibited FSCs from transitioning to follicle cells. This suggests that the relative strength of these two opposing signals helps maintain thecorrect number of FSCs while also balancing the formation of follicle and escort cells.

JAK-STAT, Wnt and other signals guide the development of many organisms, including humans, and have also been linked to cancer. Therefore, the principles and mechanisms uncovered may apply to other types of stem cells. Furthermore, this work highlights genetic changes that can allow a mutant stem cell to amplify and take over an entire stem cell community, which may play a role in cancer and other illnesses.

of a female's thirty or more ovarioles (*Figure 1A–D*) every 12 hr under optimal conditions, requiring a high constitutive rate of FC production throughout adult life (*Duhart et al., 2017*; *Margolis and Spradling, 1995*). An FC is defined by permanent association with a germline cyst and therefore passes inexorably out of the germarium within about two days and through the ovariole within five days under optimal conditions. An FSC can therefore be defined by lineage analyses as a cell that produces FCs but persists longer than an FC. However, in the original study identifying FSCs an implicit assumption was made, in accord with contemporary precedents, that each FSC is long-lived and maintained by invariant single-cell asymmetry (*Margolis and Spradling, 1995*). The consequent deductions of FSC number, location and behavior were largely re-stated as dogma over the following two decades despite some contrary observations (*Hartman et al., 2015*; *Nystul and Spradling, 2007*; *Nystul and Spradling, 2010*; *Zhang and Kalderon, 2001*). A comprehensive re-evaluation, which included the analysis of all FSC lineages, without any prior assumptions about their behavior, showed that individual FSCs were frequently lost or duplicated (*Reilein et al., 2017*) and that FSC differentiation to an FC was not temporally coupled to, or dependent upon division of the same FSC (*Reilein et al., 2018*). These characteristics of maintenance by population asymmetry, together with independent cell division and cell differentiation events and decisions, are shared by two very important and intensively studied types of mammalian epithelial stem cell, in the gut and in the epidermis (*Jones, 2010*; *Mesa et al., 2018*; *Ritsma et al., 2014*; *Rompolas et al., 2016*). The re-evaluation of FSC lineages and appreciation of population asymmetry as the governing principle not only highlighted FSCs as a suitable model for many types of mammalian stem cells but also drastically revised evaluation of the number, location and behavior of FSCs (*Reilein et al., 2017*), as summarized below.

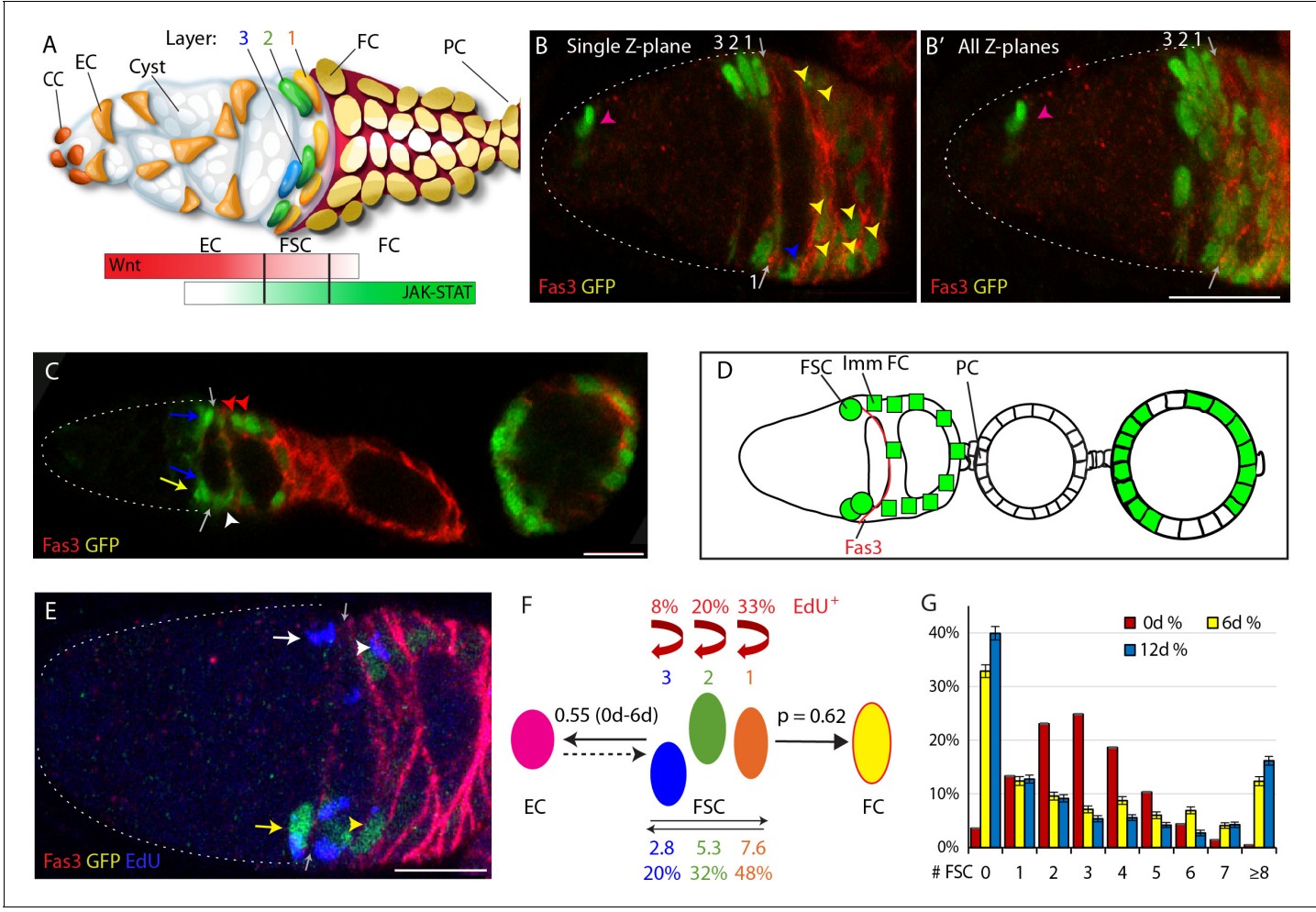

**Figure 1.** Follicle Stem Cell locations, signals and behaviors. (**A**) Cartoon representation of a germarium. Cap Cells (CC) at the anterior (left) contact Germline Stem Cells (not shown), which produce Cystoblast daughters that mature into 16 cell germline cysts (white) as they progress posteriorly. Quiescent Escort Cells (ECs) extend processes around germline cysts and support their differentiation. Follicle Stem Cells (FSCs) occupy three AP Layers (3, 2, 1) around the germarial circumference and immediately anterior to strong Fas3 staining (red) on the surface of all early Follicle Cells (FCs). FCs proliferate to form a monolayer epithelium, including specialized terminal Polar Cells (PCs), which secrete the Upd ligand responsible for generating a JAK-STAT pathway gradient (green) of opposite polarity to the Wnt pathway gradient (red), generated by ligands produced in CCs and ECs. (**B**) A GFP-positive (green) MARCM FSC lineage that includes FSCs in each layer, an EC (magenta arrowhead), a recently produced 'immediate' FC (blue arrowhead) and other FCs (yellow arrowheads) visualized together with Fas3 (red, arrows mark anterior Fas3 border) as (**B**) a single 3 μm z-section and (**B'**) a projection of ten z-sections (scoring is done by examining each z-section). The dotted white line outlines the germarium here and in future similar images. (**C, D**) Early portion of an ovariole with marked FSCs (green) in layer 1 (blue arrows) and layer 2 (yellow arrows), a marked immediate FC (white arrowhead) and more posterior FCs (magenta arrowheads) together with the anterior Fas3 (red) border (gray arrows), also (**D**) shown diagrammatically with anterior PCs of the first budded egg chamber indicated. (**E**) Germarium with a MARCM FSC lineage (green) stained for EdU (blue) incorporation during 1 hr prior to fixation, showing examples of a GFP-positive EdU+ FSC (yellow arrow) and FC (yellow arrowhead), a GFP-negative EdU+ FSC (white arrow) and FC (white arrowhead) and the anterior Fas3 (red) border (gray arrows). (**F**) Diagram showing four of five properties of FSC behavior measured for all marked FSCs in MARCM lineages, with values indicated for normal FSCs: EdU incorporation frequency for each FSC layer (red text and arrows), FSC location among the three layers (indicated by absolute numbers and frequencies), ECs produced per anterior FSC (those in layers 2 and 3) over a given period (0.55 from 0-6d), and the likelihood for a layer 1 FSC to become an FC (p=0.62) in a single budding cycle. (**G**) Distribution of the number of surviving FSCs observed for control genotypes at 6d (yellow) and 12d (blue), with the theoretically expected binomial distribution of FSCs initially marked (0d; red) based on the measured average number of surviving marked FSCs. All scale bars are 10 μm.

The online version of this article includes the following source data for figure 1:

**Source data 1.** Numerical data for graphs in *Figure 1*.

Production of 5–6 'founder' FCs (the first FCs to associate with a germline cyst to seed the FC epithelium) per budding cycle is accomplished by 14–16 FSCs, arranged in three anterior-posterior (AP) rings anterior to the border of strong Fas3 protein expression, near the mid-point of the germarium (*Figure 1A–D*; *Hayashi et al., 2020*; *Reilein et al., 2017*; *Reilein et al., 2018*). These FSCs also produce a second cell type known as an Escort Cell (EC) (*Hayashi et al., 2020*; *Reilein et al., 2017*). ECs are quiescent cells anterior to the FSC domain (*Figure 1A*) that envelop and support the differentiation of developing germline cysts (*Decotto and Spradling, 2005*; *Kirilly et al., 2011*). FCs, which first encapsulate region 2b germline cysts and are defined by continued association with a single cyst, derive directly from the posterior ('layer 1') FSCs, whereas ECs derive directly from anterior FSCs in layers 2 or 3 (*Reilein et al., 2017*).

Each FSC lineage (marked descendants of a single FSC) exhibits stochastic behaviors, including extinction or amplification and production of FCs, ECs or both. A single FSC lineage can include both ECs and FCs because FSCs can divide and can exchange AP locations over time. FSCs also exhibit extensive radial movements tracked by live imaging, and no radial germarium asymmetries are known, suggesting that all FSCs within a layer are equivalent (*Reilein et al., 2017*). Posterior FSCs divide faster than anterior FSCs, so that the roughly four-fold greater efflux of derivatives from the posterior face of the FSC domain is supported without significant net flow of FSCs from anterior to posterior locations (*Reilein et al., 2017*; *Reilein et al., 2018*). Thus, FSCs present a paradigm of dynamic heterogeneity (*Greulich and Simons, 2016*), where each stem cell within a fluid spatially-defined community exhibits distinctive instantaneous properties characteristic of its precise AP location but future behavior can change as a result of apparently stochastic changes in position or cell division. How are these heterogeneous individual cell behaviors marshaled into a defined stem cell domain that maintains a roughly constant number of stem cells and a continuous supply of an appropriate number of FC and EC products?

FSC maintenance and amplification have been found to depend on the activity of many of the major pathways initiated by extracellular signals. The earliest studies highlighted the role of Hedgehog (Hh) signaling (*Zhang and Kalderon, 2001*). Hh is produced in Terminal Filament and Cap cells, the anteriormost cells of the germarium, and its release is regulated by the Hedgehog binding protein Boi (*Forbes et al., 1996a*; *Hartman et al., 2010*). Hh was shown to promote FSC survival and amplification principally by regulating the rate of FSC division through transcriptional induction of the Hippo-pathway transcriptional co-activator Yorkie (Yki) (*Huang and Kalderon, 2014*). This constitutive role of Hh signaling in well-fed flies is also part of an environmental sensor, with nutrient deprivation leading to reduced Hh dispersal and consequent slowing of FC and egg chamber production (*Hartman et al., 2013*). The key role of FSC division rate for FSC competition was highlighted by studies of Hh signaling and also by the discovery of several regulators of proliferation in a genetic screen for FSC maintenance factors (*Wang et al., 2012*; *Wang and Kalderon, 2009*). A functional connection between stem cell division rate and competition is not expected for stem cells maintained by invariant single-cell asymmetry and was finally explained by the finding that FSC differentiation is independent of FSC division (*Reilein et al., 2018*).

Although the Hh signal is graded, declining from anterior to posterior, initial tests indicated that graded signaling was not important for continued normal FSC function (*Vied et al., 2012*). BMP, EGF, integrin and insulin receptor initiated pathways have also been implicated in FSC function (*Castanieto et al., 2014*; *Johnston et al., 2016*; *Kirilly et al., 2005*; *O'Reilly et al., 2008*; *Vied et al., 2012*; *Wang et al., 2012*) but the two pathways that have emerged so far as the strongest candidates for determining niche space and position-specific stem cell behaviors are the Wnt and JAK-STAT pathways because they both have graded activities in the AP dimension (*Figure 1A*; *Reilein et al., 2017*; *Vied et al., 2012*; *Wang and Page-McCaw, 2014*) and they both have a very strong influence on FSC behavior (*Reilein et al., 2017*; *Song and Xie, 2003*; *Vied et al., 2012*).

The Wg and Wnt6 ligands are produced in Cap Cells at the anterior of the germarium and are supplemented by the production of Wnt2 and Wnt4 in ECs (*Forbes et al., 1996b*; *Luo et al., 2015*; *Sahai-Hernandez and Nystul, 2013*; *Waghmare et al., 2018*; *Wang and Page-McCaw, 2018*) to produce high levels of pathway activity over the EC domain with a sharp decline over the FSC domain and little or no activity in FCs (*Figure 1A*; *Reilein et al., 2017*; *Wang and Page-McCaw, 2014*). FSCs were lost from the niche cell autonomously when Wnt signaling was either genetically elevated or reduced (*Song and Xie, 2003*; *Vied et al., 2012*). More recently it was shown that the

primary effects of altering Wnt pathway activity were exerted on the AP location of FSCs and their conversion to differentiated products, with increased pathway activity favoring more anterior locations and EC production, while reducing FC production (*Reilein et al., 2017*). Thus, relatively rapid loss of FSCs due to elevated Wnt pathway activity results from conversion of all FSCs over time to ECs.

The JAK-STAT ligand Unpaired (Upd) is produced in specialized FCs called polar cells that are found at the anterior and posterior ends of developing egg chambers (*Figure 1A, D*; *McGregor et al., 2002*; *Vied et al., 2012*). Pathway activity is high in FCs in the germarium and decreases from posterior to anterior over the FSC domain with only low levels in ECs (*Figure 1A*; *Vied et al., 2012*). When JAK-STAT activity was elevated in FSC lineages, it was shown that these FSCs out-competed wild-type FSCs and that proliferating FSC derivatives could accumulate in EC territory. Conversely, loss of STAT activity resulted in accelerated FSC loss (*Vied et al., 2012*). Those studies suggested potential roles in both FSC division and location or differentiation but detailed analysis was not possible at that time, before understanding of the organization and behavior of FSCs was drastically revised (*Reilein et al., 2017*).

Here, we have dissected cell autonomous responses to genetic changes in Wnt and JAK-STAT signaling pathways to separate their influences on each potentially independently controlled and separately measured parameter of FSC behavior: (1) FSC division rates, (2) FSC AP location, (3) FSC conversion to FCs, and (4) FSC conversion to ECs, which combine to control FSC competitive status, measured by changes in FSC numbers over time (*Figure 1F*). The results showed that these two graded pathways are substantially responsible for defining the FSC domain. The polarity and consequent magnitude of each graded pathway activity promoted differentiation to ECs at the anterior and differentiation to FCs at the posterior of the FSC domain, with especially sensitive responses to Wnt at the anterior and to JAK-STAT at the posterior. The magnitude of JAK-STAT signaling also substantially influenced the spatial pattern of cell divisions. Some co-ordination of FSC division and differentiation results from the dual role of the JAK-STAT pathway in promoting FSC division and FSC conversion to FCs but this coordination did not suffice in the artificial absence of Wnt signaling. Finally, the general correspondence between overall FSC competitive outcomes and the independent constituent behaviors (division rate, AP location, and differentiation rate to FCs or ECs) measured under a large variety of genetic conditions provides further support for the current view of FSC numbers, locations and behaviors (*Reilein et al., 2017*; *Reilein et al., 2018*).

## Results

### Cell lineage approach to measure five separable parameters of FSC behavior

Prior to 2017, when each ovariole was thought to harbor just two or three FSCs, the cell autonomous effects of altered genotypes on FSC biology were ascertained by measuring the frequency of surviving marked FSC clones, defined by the presence of labeled FCs and a putative FSC, at various times after clone induction relative to control genotypes tested in parallel (*Castanieto et al., 2014*; *Kirilly et al., 2005*; *O'Reilly et al., 2008*; *Song and Xie, 2003*; *Vied et al., 2012*; *Wang et al., 2012*). Numerous genetic changes were found to reduce FSC clone survival severely. Occasionally, the normally low frequency of ovarioles containing only marked FSCs and FCs ('all-marked') was also elevated, indicating a genotype that markedly increased FSC competitiveness. Now that it is appreciated that there are 14–16 FSCs in distinct AP locations, associated with different instantaneous division rates and differentiation potential, and that FSCs produce EC as well as FCs (*Hayashi et al., 2020*; *Reilein et al., 2017*; *Waghmare et al., 2018*), the results of clonal analysis can reveal far more about the effect of a specific genotype on different aspects of FSC behavior and the net effect on FSC survival and amplification can be measured more precisely by measuring FSC numbers. Correspondingly, labeled lineages must be scored in far more detail than before to reveal that information.

We conducted an extensive series of experiments using a standard regime in order to extract comprehensive quantitative information about FSC behavior and to be able to compare results for a large number of altered genotypes among all experiments in the series. We induced GFP-labeled clones in dividing cells of young, well-fed adult females using the MARCM (Mosaic Analysis with a

Repressible Cell Marker) system (*Lee and Luo, 2001*) with constitutive drivers (*actin-GAL4* and *tubulin-GAL4* together) of *UAS-GFP* and, where relevant, additional transgene expression. Heat-shock induction of a *hs-flp* recombinase transgene elicited recombination at the base of the relevant chromosome arm (using *FRT* recombination sites on 2L, 2R or 3R) in a fraction of FSCs (about 20%) to create homozygous recessive mutations or activate expression of a transgene (or both). After 6 or 12 days, ovarioles were dissected, labeled for 1 hr with the nucleotide analog EdU to measure cell division (for the 6d test only), fixed and stained to label all nuclei and the cell surface protein, Fasciclin 3 (Fas3). Each experiment included a variety of altered genotypes and a control with the same *FRT* recombination site. For each sample, GFP-labeled FC locations along the ovariole were recorded (*Figure 1C,D*) and complete confocal z-section stacks of the germarium were archived and analyzed to count labeled FSCs in layer 1, immediately anterior to the border of strong Fas3 staining, and in the next two anterior layers (2 and 3), as well as labeled ECs (anterior to FSCs) (*Figure 1B*), scoring also the number of labeled ECs and FSCs that had incorporated EdU (at 6d) (*Figure 1E*).

Our objective was to use the results to measure each distinguishable parameter of FSC behavior or decision-making separately (*Figure 1F*). Cell division over the FSC domain was measured by EdU incorporation at the earlier time-point (6d) so that sufficient FSCs of poorly competitive genotypes were still present in good numbers and hyper-competitive FSCs were not sufficiently abundant to potentially distort germarial morphology or induce secondary, non-autonomous responses. Genotype-dependent changes in the precise AP location of FSCs within the FSC domain were evident at 6d but were consistently most prominent at 12d, as were changes in the average number of FSCs present, so only the 12d results are presented. FC production was assessed quantitatively by a method we devised for measuring the probability of FC production per posterior FSC in one cycle of egg chamber budding ('p' in *Figure 1F*), using 6d samples to ensure a suitably low frequency of posterior FSCs for all genotypes. EC production was measured from 0-6d and 0-12d; it was normalized to the inferred number of anterior FSCs present during those periods.

Normal FSC behavior was reported by controls from 31 separate MARCM experiments, scoring at least 50 ovarioles in almost every case. The results were extremely similar to those deduced previously from the more limited set of multicolor and MARCM experiments that formed the basis of our current perception of FSCs (*Reilein et al., 2017*). The results for controls are summarized in *Figure 1F* and will be referenced individually later, in the context of genetic changes that alter those behaviors.

Here we note that each germarium contained an average of 3.2 marked FSCs at 6d and 3.3 marked FSCs at 12d, counting all ovarioles, including those with no labeled cells. If marked FSCs of the control genotype have no competitive advantage or disadvantage over unmarked FSCs it is expected that the average number of labeled FSCs should remain constant, as observed, and these measurements should therefore report the average number of FSCs initially labeled in each germarium (as 3.2–3.3 at 0d). When deducing FSC properties for the first time it was often important to assay ovarioles with lineages derived from a single FSC (*Fox et al., 2008*; *Kretzschmar and Watt, 2012*; *Reilein et al., 2017*). For example, FSC lineages with only a single candidate FSC at the time of examination were used to ascertain the location of FSCs as being in any radial location, most frequently in layer one or in layer two and occasionally in layer three (*Reilein et al., 2017*). Similarly, the ability of an FSC to produce both FCs and ECs was demonstrated by generating FSC lineages at very low frequency, so that most lineages originated from a single cell (*Reilein et al., 2017*). In the present studies we already can define FSCs by location and the labeling of over three FSCs per ovariole is advantageous because it effectively allows us to examine the fate of a larger number of FSCs for a given number of ovarioles. Initial labeling of each FSC in a germarium is theoretically independent and the chance of an FSC being labeled in any two germaria is theoretically equal, so the number of initially labeled FSCs per ovariole can be estimated to have a binomial distribution centered around 3.2–3.3 ('0d', red in *Figure 1G*). The observed distribution at 6d was quite different from the assumed starting distribution, most obviously because more than a third of ovarioles no longer included any FSCs, while the proportion of ovarioles with six or more FSCs had increased (*Figure 1G*), consistent with expectations for neutral competition (*Jones, 2010*; *Reilein et al., 2017*; *Reilein et al., 2018*). These changes were further exaggerated at 12d but full colonization of a germarium by marked cells generally takes longer (*Reilein et al., 2017*), so even at 12d marked FSCs remain in a minority and are competing against unmarked wild-type cells in almost all ovarioles

(*Figure 1G*). That circumstance also applies to almost all variant genotypes investigated, ensuring that results reflect competition of marked FSCs with wild-type FSCs.

## Graded JAK-STAT signaling instructs graded FSC proliferation

In control clones 6d after induction, the average percentage of all GFP-marked FSCs that incorporated EdU was 25.1% (n = 4753) with a pronounced gradient of labeling, declining from posterior to anterior (33.4% for layer 1, 20.0% for layer 2, and 8.2% for layer 3) (*Figure 2A*), similar to previous observations (*Reilein et al., 2017*). It had previously been observed that FSCs are rapidly lost in the absence of STAT activity, and that FSCs with excess JAK-STAT pathway activity became unusually numerous and included derivatives that incorporated EdU within the EC domain, suggesting that this pathway may affect FSC proliferation (*Vied et al., 2012*). However, the quantitative effect of JAK-STAT signaling on FSC division rates has not been reported. We found that only 2.4% of FSCs with either of two homozygous null *stat* alleles incorporated EdU (n = 336), a ten-fold reduction compared to controls (*Figure 2A,B*). To increase JAK-STAT activity, we expressed excess levels of the only *Drosophila* Janus Kinase, Hopscotch (Hop) using a *UAS-Hop* transgene in FSC clones (*Vied et al., 2012*; *Xi et al., 2003*), and found that 43.9% of *UAS-Hop* FSCs incorporated EdU (n = 1379), nearly double the rate of control FSC clones (*Figure 2A,C*; *Figure 2—figure supplement 1*). The pattern of EdU incorporation in these FSCs still showed a posterior bias, with 49.9% of layer 1, 39.8% of layer 2, and 32.1% of layer 3 *UAS-Hop* FSCs labeled by EdU, although the EdU indices of layer 1 and layer 3 FSCs relative to the whole FSC population were significantly different from controls (*Figure 2A*). These experiments demonstrated that the JAK-STAT pathway has a very strong positive, dose-responsive, cell autonomous influence on the FSC cell cycle. We also saw that labeled cells in the EC region, which normally do not divide at all, sometimes incorporated EdU (12.4%) when JAK-STAT pathway activity was elevated (*Figure 2A*; *Figure 2—figure supplement 1B*), as noted previously without quantitation (*Vied et al., 2012*).

JAK-STAT pathway activity, reported by a 'STAT-GFP' transgene with ten tandem STAT binding sites (*Bach et al., 2007*), is graded from posterior to anterior over the FSC domain (*Figure 2D,F*), with a major ligand emanating from polar follicle cells (*Vied et al., 2012*). Because the JAK-STAT activity gradient runs parallel to the graded pattern of EdU labeling we wished to test whether the two gradients were causally related. To do this, we took advantage of the *C587-GAL4* driver, which is expressed strongly in the anterior of the germarium and decreases in strength towards the posterior with almost no detectable expression in FCs (*Reilein et al., 2017*; *Song et al., 2004*). This pattern is roughly a mirror-image of the normal JAK-STAT signaling pathway gradient. We expressed *UAS-Hop* from the *C587-GAL4* driver (*C587>Hop*), utilizing a temperature-sensitive *GAL80* transgene (*Zeidler et al., 2004*) to restrict *UAS-Hop* expression temporally. After 3d at the restrictive temperature of 29C, we measured STAT-GFP fluorescence and found it to be similar in each of the three FSC layers and also over more anterior regions, indicating that the entire FSC domain now has roughly even JAK-STAT pathway activity (*Figure 2E,F*).

With roughly even JAK-STAT activity across the FSC region, we measured proliferation in the three FSC layers. We observed nearly identical frequencies of EdU labeling in each FSC layer of *C587>Hop* germaria; 38.2% of layer 1, 37.4% of layer 2, and 41.1% of layer 3 FSCs (*Figure 2G,H,J*). Thus, synthetically making JAK-STAT pathway activity uniform, rather than graded, eliminated the normal posterior to anterior gradient of EdU labeling.

Additionally, elevated JAK-STAT activity in the anterior of the germarium stimulated EdU incorporation in 10.4% of ECs (*Figure 2H,J*). In this experiment, those cells were quiescent ECs prior to increasing JAK-STAT activity with *C587-GAL4* and temperature elevation. In the MARCM studies, the GFP-marked dividing cells in the EC region originated instead from marked FSCs with elevated JAK-STAT signaling. Clearly, excess JAK-STAT pathway activity can suffice to initiate cell division in the EC domain, whether the target cells were recently derived from FSCs or not. The rate of division of those cells, indicated by their EdU index, was substantially lower than for cells in the FSC domain (*Figure 2J*) despite similar levels of JAK-STAT pathway activity in the *C587-GAL4/UAS-Hop* experiment (*Figure 2F*), suggesting the presence of other factors restricting EC division or inertia due to prior quiescence (*Spencer et al., 2013*).

For comparison, we tested the effect of increasing CycE expression. In MARCM clones expressing *UAS-CycE* the FSC EdU index was greatly increased (*Figure 2K*) but we observed no EdU incorporation in the EC region. When *UAS-CycE* was expressed with the *C587-GAL4* driver (*C587>CycE*) we

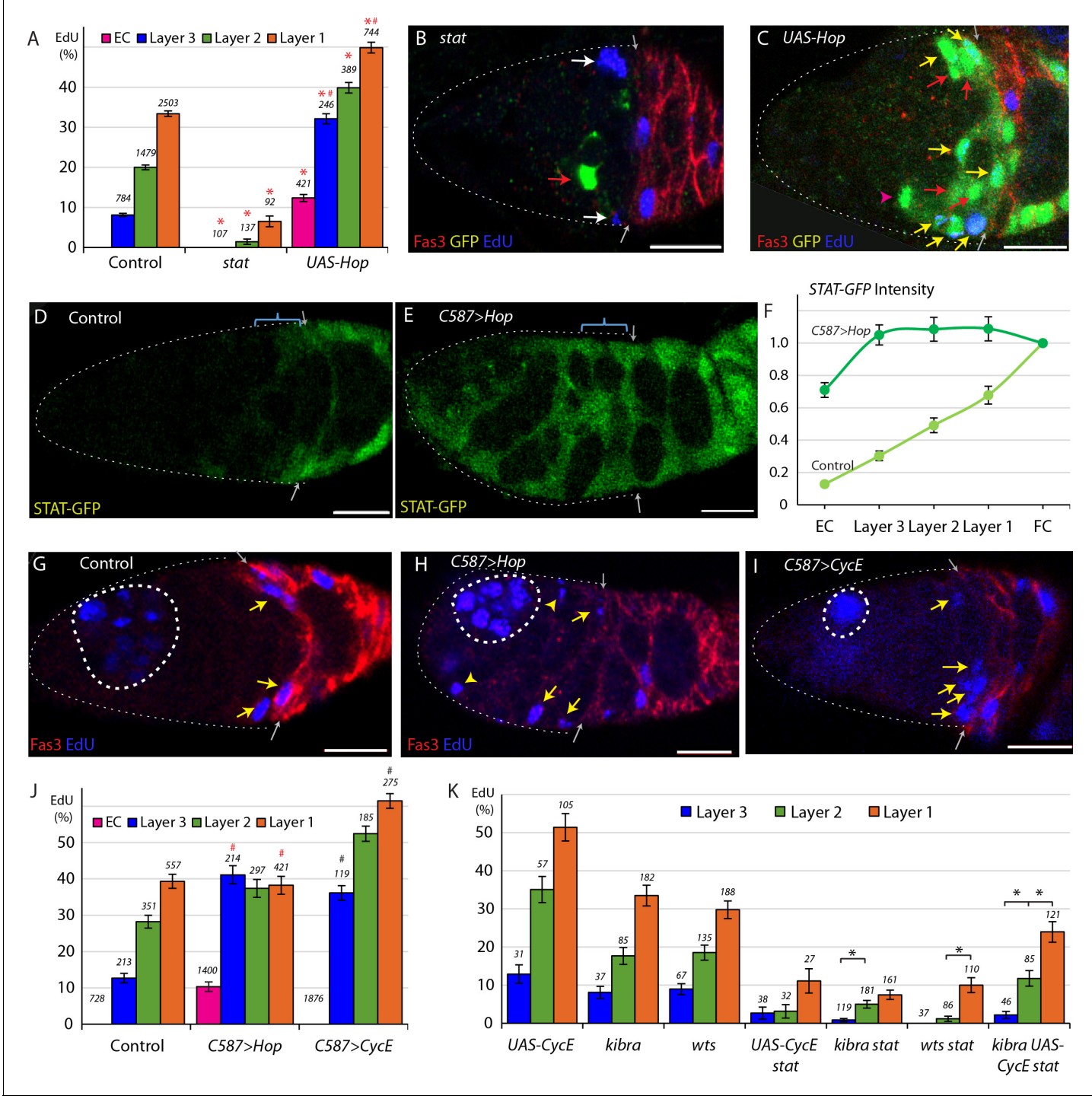

**Figure 2.** Graded JAK-STAT signaling determines FSC and EC proliferation profile. (A) EdU incorporation frequency into FSCs in layers 1–3 and ECs for the indicated genotypes of MARCM lineages, with the number of cells scored and significant differences to control values (red asterisks, p<0.001). The statistical significance of the EdU index of each FSC layer as a fraction of overall EdU index relative to controls was calculated for all genotypes. Significant differences are indicated by the # symbol (# [in red] p<0.001. (B, C) EdU (blue) incorporation into MARCM FSC lineages 6d after clone induction with anterior border of Fas3 (red) marked (gray arrows). (B) Most FSCs lacking *stat* activity (green) did not incorporate EdU (red arrow), unlike unmarked GFP-negative neighbors (white arrows). (C) Increased JAK-STAT pathway activity from expression of *UAS-Hop* produced many GFP-positive EdU⁺ FSCs (yellow arrows). GFP-positive EdU⁻ FSCs (red arrows) and a GFP-positive EdU⁻ EC (magenta arrowhead) are also indicated. (D–E) *STAT-GFP* reporter activity (green) (D) normally declines from the posterior over the FSC domain (blue bracket; arrows mark Fas3 (not shown) anterior border) but (E) becomes uniformly high 3d after *UAS-Hop* expression with *C587-GAL4*. (F) Average relative intensity of GFP fluorescence from STAT-GFP reporter in

*Figure 2 continued on next page*

*Figure 2 continued*

the indicated cell types for Control (n = 20) and *C587 >Hop* (n = 22) germaria. (G–I) EdU (blue) incorporation in somatic cells is (G) normally restricted to FSCs (yellow arrows) and FCs beyond the Fas3 (red) border (gray arrows), (I) even when CycE activity is increased, but (H) ECs (arrowheads) are also labeled when JAK-STAT pathway activity is elevated. Thick dashed white lines outline germline cysts labeled by EdU. Thin dotted white line outlines the germarium. (J) EdU incorporation frequency into FSCs of layers 1–3 and ECs for germaria with the indicated genotypes. EdU incorporation is expressed as the percentage of all counted FSCs in layers 1, 2 and 3, while the total number of ECs in each germarium was not counted but assumed to be 40 in all cases (the total number of DAPI-labeled nuclei scored is above each column). The statistical significance of the EdU index of each FSC layer as a fraction of overall EdU index relative to controls was calculated for both altered genotypes. Significant differences are indicated by the # symbol (# p<0.05, # [in red] p<0.001). (K) EdU incorporation frequency into FSCs of layers 1–3 and ECs for the indicated genotypes of MARCM lineages with number of cells scored above each column and significant differences between pairs of FSC layers indicated only for *stat*-containing genotypes (black asterisks, p<0.05). The statistical significance of the EdU index of each FSC layer as a fraction of overall EdU index relative to controls was also calculated for all genotypes. No significant differences were found, indicating a gradient of EdU incorporation that is not significantly different from controls. All scale bars are 10 µm. See also *Figure 2—figure supplement 1*.

The online version of this article includes the following source data and figure supplement(s) for figure 2:

**Source data 1.** Numerical data for graphs in *Figure 2*.
**Figure supplement 1.** JAK-STAT pathway promotes division of FSCs and cells in EC territory.

found that the profile of EdU incorporation for FSCs remained graded with a posterior basis, contrasting with the response to *UAS-Hop*, although the gradient was flatter than in control FSCs, as revealed by statistically significant differences in the relative EdU index for layer one and for layer three relative to the overall EdU index (*Figure 2I,J*). Also, ECs remained quiescent. We conclude that graded JAK-STAT pathway activity instructs graded proliferation within the FSC domain, and that the anterior range of sufficient JAK-STAT pathway activity appears to define the anterior boundary of this critical stem cell property.

## JAK-STAT is not the sole potential contributor to the FSC proliferation gradient

If JAK-STAT signaling were uniquely responsible for regulating the FSC proliferation gradient, then we would not expect there to be any bias in EdU incorporation, by layer, for FSC MARCM clones that have no JAK-STAT pathway activity. Though EdU incorporation was very low in *stat* FSC clones, it was graded; 6.5% of marked layer 1 FSCs incorporated EdU, compared to 1.5% for layer 2 and 0% for layer 3 (*Figure 2A*). Thus, there appear to be other influences that pattern FSC proliferation. Their magnitude is, however, hard to assess reliably from this experiment alone because *stat* mutant FSC cycling is very infrequent.

We therefore sought to introduce additional genetic changes onto a *stat* mutant background that would increase FSC proliferation in the marked MARCM lineages without themselves altering the normal graded pattern of proliferation. We found that expression of excess CycE and inactivation of upstream components of the Hippo/Yorkie pathway, Kibra and Warts (Wts), previously shown to influence FSC proliferation (*Huang and Kalderon, 2014*), appear to fulfill this requirement because the gradient of EdU labeling was largely unaltered (*Figure 2K*). When each of these three manipulations was paired with *stat*, the average frequency of EdU labeling roughly doubled compared to *stat* alone (2.4%), with 4.8% of *kibra stat* FSCs (n = 461), 5.2% of *wts stat* FSCs (n = 231), and 5.2% of *stat UAS-CycE* FSCs (n = 97) incorporating EdU, while *kibra* and *UAS-CycE* together increased EdU incorporation to 15.9% (n = 252) (*Figure 2K*). In all of these experiments, more FSCs in layer one incorporated EdU compared to the anterior layers in a pattern resembling that of normal FSCs (*Figure 2K*). Moreover, there were no statistically significant differences from controls in the relative EdU index of any one layer relative to the whole FSC population for these genotypes (*Figure 2K*) or *stat* alone (*Figure 2A*), indicating that, in the absence of graded JAK-STAT pathway activity in the marked cells, there remains a robust mechanism for imposing graded FSC proliferation. Once the source of this mechanism is identified it will be possible to test whether it contributes to graded FSC proliferation under normal conditions or is effective only in the absence of JAK-STAT pathway activity.

## JAK-STAT pathway activity opposes anterior accumulation of FSCs

When scoring germaria with *stat* mutant clones, we observed that 96.5% of *stat* FSCs were found in the anterior layers of the germarium by 12d (*Figure 3A*). Since loss of STAT drastically reduces FSC proliferation we considered whether the location of FSCs might depend on their division rate. For example, even though there is exchange of FSCs between layers (*Reilein et al., 2017*), a proliferation-deficient FSC may compete less well in layer 1. When we tested other mutants that had severely impaired proliferation, including *cycE* and *cutlet* (*Wang et al., 2012*; *Wang and Kalderon, 2009*), we observed an altered distribution of FSCs with the proportion of FSCs in layer one reduced from a control value of 48.8% to 41.0% for *cycE*$^{WX}$ hypomorphs and 43.3% for *cutlet* FSCs but those changes were not statistically significant and were much smaller than observed for *stat* mutant FSCs (*Figure 3A*).

We also tested the consequences of increasing the division rate of *stat* mutant FSCs using the *kibra, wts,* and *UAS-CycE* manipulations. The proportion of layer 1 FSCs was 29.7% for *kibra stat, wts stat*, and *UAS-CycE stat* FSCs ('*kibra/wts/UAS-CycE stat*'; average for the three aggregated genotypes), and 36.5% for *kibra UAS-CycE stat* FSCs, both significantly lower than for controls (49%) but higher than for *stat* alone (3.5%) (*Figure 3A,D,E*). Thus, we observed a consistent anterior bias for all FSCs lacking STAT activity, even for genotypes that permitted EdU incorporation at frequencies approaching normal values. The observation that *stat* mutant FSCs had a reduced anterior bias when their proliferation was enhanced also supports the hypothesis that reduced division rates selectively deplete FSCs from the fastest-dividing, posterior layer.

## JAK- STAT pathway activity promotes FC production from posterior FSCs; role of Fas3

The signals and mechanisms that govern conversion of an FSC to an FC are largely unknown. In fact, only with recent insights into FSC organization can we measure this process independently of other factors, such as altered division rates, in order to attribute changes in FSC survival to changes in the frequency of differentiation to FCs (*Figure 1F*). Importantly, an FSC can become an FC at any time relative to its last division, and FSC division and differentiation can therefore potentially be regulated independently (*Reilein et al., 2018*). By correlating the location of labeled FSCs with recent production of FCs it was determined that all or most FCs derive directly from layer 1; in other words, only layer 1 FSCs associate with a passing germline cyst to become an FC (*Reilein et al., 2017*). Previous studies also found that a single founder FC produced a patch occupying 17.8% of the monolayer of an egg chamber on average, which translates to an average of 5.6 founder FCs (1/0.178) produced per cycle of egg chamber budding (*Reilein et al., 2018*).

We devised a method to determine the probability that a single FSC becomes an FC in one cycle of FC recruitment from our MARCM data. The germarium generally includes one stage 2b cyst and one stage three cyst contacting Fas3-positive FCs (*Figure 1C,D*). We call the first layer of Fas3-positive cells adjacent to the posterior face of the stage 2b germline cyst 'immediate FCs' to acknowledge that these cells very likely became FCs during the most recent cycle of FC allocation to a cyst (*Figure 1B–D*). The designation of 'immediate FCs' reflects a location used for scoring, with no implication of specialized properties. We can reliably score if there is no labeled immediate FC in a germarium but we cannot reliably score the number of immediate FCs. We therefore scored the presence or absence of immediate FCs in germaria with 0–3 marked posterior FSCs in 6d samples. Germaria with higher numbers of marked layer 1 FSCs almost invariably include marked immediate FCs (layer 1 FSCs become FCs at a high frequency) and are therefore not informative for calculating the frequency of conversion of a single layer 1 FSC to an FC. From 556 control germaria, across 31 MARCM experiments, we found that a layer 1 FSC has, on average, a 61.6% likelihood (p=0.616) of becoming an FC in one cycle (see Materials and methods) (*Figure 3C*).

The calculation method assumes that 7 of 16 FSCs divide in an average budding cycle, based on data from *Reilein et al., 2018*, so that the number of layer 1 FSCs available for conversion to FCs in one cycle is higher than the measured steady-state number by a factor of a half (because new FSCs will only be present on average for half the cycle) of 7/16. The average number of layer 1 FSCs measured from 31 control experiments was 7.6 (with 5.3 in layer 2 and 2.8 in layer 3; *Figure 1F*), so the expected yield of FCs per cycle is 7.6 (1+ 7/32)(0.616)=6.1. The result is close to the value of 5.6

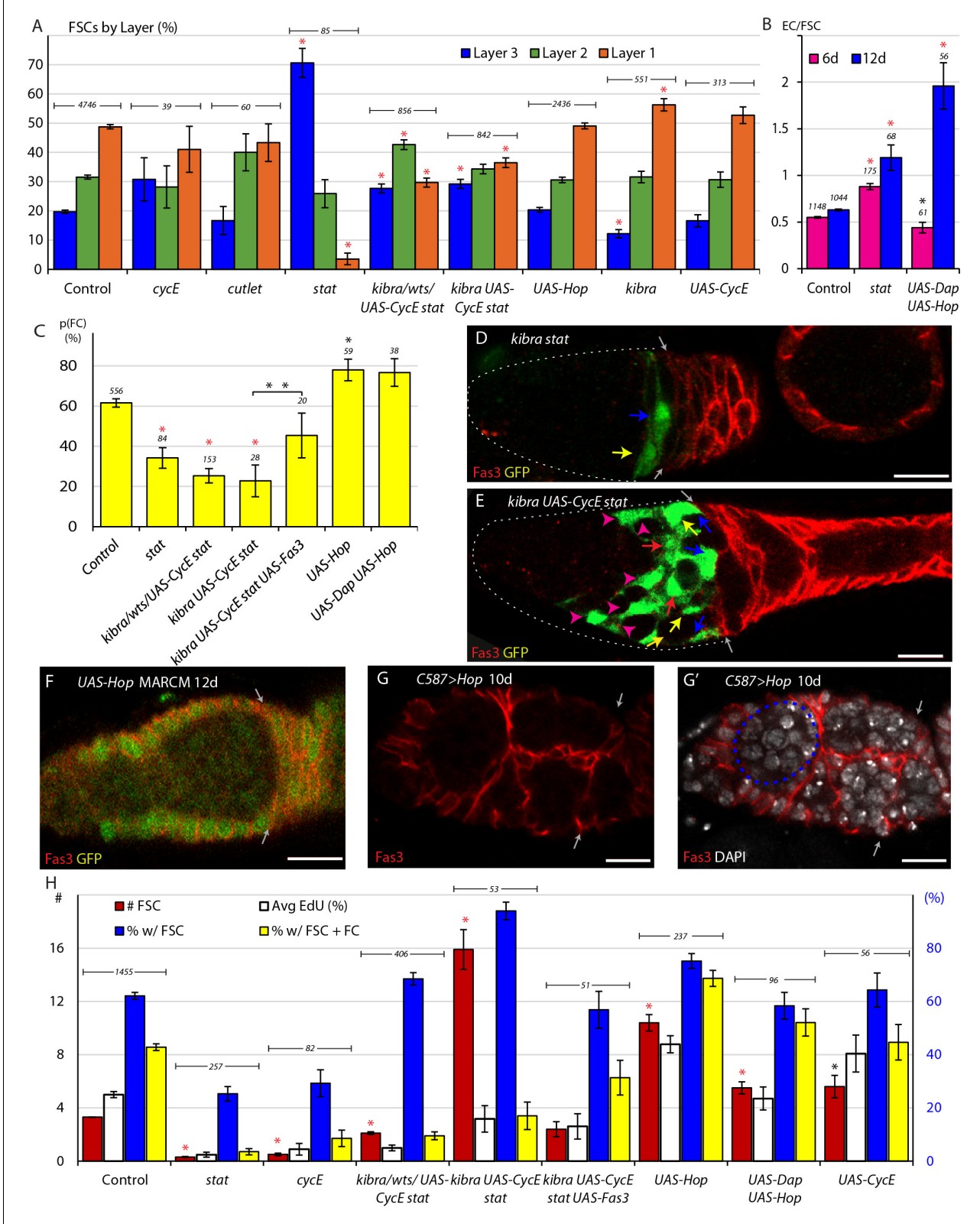

**Figure 3.** JAK-STAT signaling promotes conversion of FSCs to FCs. (**A**) Relative frequency of marked FSCs in each layer for the indicated genotypes of MARCM lineages, with the number of total FSCs scored for each genotype and significant differences to control values (red asterisks, p<0.001). Here and elsewhere, '*kibra/wts/UAS-CycE stat*' is the sum of all tests with *kibra stat*, *wts stat* and *UAS-CycE stat*. (**B**) ECs produced per anterior FSC from 0-6d (magenta) and 0-12d (blue) for the indicated MARCM lineage genotypes with the total number of relevant germaria scored and significant

*Figure 3 continued on next page*

*Figure 3 continued*

differences to control values (black asterisk, p<0.05, red asterisks, p<0.001). (**C**) Average probability of a layer 1 FSC becoming an FC during a single budding cycle for the indicated MARCM lineage genotypes with the number of informative germaria scored and significant differences to control values, or for the bracketed comparison (double asterisks) showing the impact of *UAS-Fas3* (black asterisks, p<0.05, red asterisks, p<0.001). (**D–E**) Despite marked FSCs in layer 1 (blue arrows), and a significant number of layer 2 FSCs (yellow arrows), layer 3 FSCs (red arrows), and ECs (magenta arrowheads), marked FCs, posterior to the Fas3 (red) border (gray arrows) are absent here (and were generally rare) in *kibra stat* and *kibra UAS-CycE stat* MARCM lineages. (**F–G**) Increased JAK-STAT pathway activity induced ectopic anterior Fas3 (red) expression (**F**) cell autonomously in MARCM lineages (green) and (**G–G'**) in numerous cells anterior to the normal Fas3 border (arrows) when increased throughout the anterior germarium using *C587-GAL4*, sometimes partitioning single cysts, visualized by DAPI (white) nuclear staining, into egg chamber-like structures (dashed blue line). (**H**) Number of FSCs per germarium (red) using y-axis scale on the left, percentage of FSCs incorporating EdU (aggregating all layers, white), percentage of ovarioles with a marked FSC (blue) and percentage of ovarioles with a marked FSC and marked FCs (yellow) (percentage y-axis scale in blue on the right) for the indicated genotypes, with the number of germaria scored at 12d (EdU was scored at 6d with n reported in (**A**)) and significant differences for FSC numbers compared to control values (black asterisk, p<0.05, red asterisks, p<0.001). All scale bars are 10 μm.

The online version of this article includes the following source data for figure 3:

**Source data 1.** Numerical data for graphs in *Figure 3*.

calculated from founder FC clone sizes, validating the method employed to calculate the probability of FC formation per FSC.

When applying the same method to mutant genotypes, the calculations factored in measured changes in FSC division rate relative to controls because that influences the total number of FSCs available for conversion to FCs during a cycle (see Materials and methods). Using this method, we found that a *stat* mutant layer 1 FSC was much less likely than controls (34.2% compared to 61.6%) to produce an FC (*Figure 3C*). To determine if reduced FC production was dependent on FSC division rate we looked at *kibra stat, wts stat,* and *UAS-CycE stat* genotypes,. The average likelihood of an FSC becoming an FC was 25.3% for these three genotypes (aggregated), and it was 22.8% for *kibra UAS-CycE stat* mutant FSCs (*Figure 3C–E*). These experiments demonstrated that FC production from its immediate precursor, a layer 1 FSC, is greatly impaired in the absence of STAT activity and that this reduction is not related to changes in the rate of FSC division.

We also examined the consequences of increasing JAK-STAT pathway activity in a layer 1 FSC. We found that the probability of becoming an FC increased from 61.6% to 78.1% for FSCs expressing *UAS-Hop* (*Figure 3C*). To test any contribution of altered FSC division we introduced a *UAS-Dacapo* (*UAS-Dap*) transgene, encoding a CycE/Cdk2 inhibitor (*Lane et al., 1996*; *Lehner et al., 1992*). We found that 23.5% of *UAS-Dap UAS-Hop* FSCs incorporated EdU, a frequency similar to controls (*Figure 3H*). Layer 1 FSCs expressing both *UAS-Hop* and *UAS-Dap* also had a higher probability than controls of becoming FCs (76.7% vs 61.6%) (*Figure 3C*). Thus, both increased and decreased JAK-STAT pathway activity significantly affected the production of FCs from FSCs independent of FSC division rate, suggesting that the magnitude of pathway activity is an important factor in regulating this transition.

The proportion of FSCs in layer one depends not only on movements between FSC layers but also on the rate of depletion from layer one to form FCs. Loss of STAT activity in multiple genotypes reduced layer one occupancy even though conversion of layer 1 FSCs to FCs was reduced, suggesting that the bias towards anterior movement within the FSC domain is even stronger than measured simply by steady-state AP distribution (*Figure 3A*). Excess JAK-STAT pathway did not affect steady-state AP location but increased conversion of layer 1 FSCs to FCs, suggesting that there is in fact a bias towards posterior movement within the FSC domain that matches the increased conversion of layer 1 FSCs to FCs. Thus, both FSC flux into layer one and conversion of layer 1 FSCs to FCs are enhanced by increased JAK-STAT signaling and opposed by loss of JAK-STAT pathway activity.

It was previously noted that strong expression of the surface adhesion molecule Fas3, which is normally observed only in FCs, was induced in some derivatives of FSCs with elevated JAK-STAT signaling in the EC and FSC domains (*Vied et al., 2012*). We confirmed these observations for *UAS-Hop* MARCM clones, finding that 66% of germaria with labeled cells in the anterior half of the germarium (the FSC and EC domains) showed ectopic Fas3 expression at 12d after clone induction (*Figure 3F*). Furthermore, we observed that these cells sometimes appeared to form a crude epithelial monolayer surrounding developing germline cysts, indicative of FC behavior. Similar structures were observed in germaria where *UAS-Hop* was conditionally expressed using *C587-GAL4*

(*Figure 3G*). Here, 53% of germaria included some cells with ectopic Fas3 expression by 3d, increasing to 72% by 6d and 94% by 10d. Thus, high JAK-STAT pathway alone can instruct at least some aspects of the FC phenotype even in locations where FCs do not normally form.

To test whether Fas3 might be an important intermediate in the normal influence of JAK-STAT signaling on FC production we expressed excess Fas3 in *kibra UAS-CycE stat* mutant FSCs. We observed a doubling in layer 1 FSC to FC conversion (from 22.8% to 45.4%) (*Figure 3C*), suggesting that increased Fas3 expression can partially restore FC production in the absence of JAK-STAT pathway activity. The mechanisms controlling Fas3 expression and its role in supporting FC production remain to be explored more fully.

## Net effect of JAK-STAT on cell-autonomous FSC longevity and amplification

By measuring the impact of altered genotypes on each component of FSC behavior (*Figure 1F*) it should be possible to predict, or at least rationalize, the net effect on FSC competitive behavior in MARCM lineage analyses, measured by the proportion of ovarioles that retain marked FSCs over time, or measured more precisely by the average number of marked FSCs per ovariole (counting all ovarioles).

For FSCs and other stem cells governed by population asymmetry in which differentiation is independent of stem cell division, the rate of stem cell division is necessarily a major determinant of competitive success (*Reilein et al., 2018*). Excess JAK-STAT pathway activity substantially increased the FSC EdU index. Accordingly, by 12d, there were an average of 10.4 *UAS-Hop* FSCs per germarium (counting all ovarioles), significantly greater than the 3.3 FSCs per germarium observed in controls (*Figure 3H*). The proportion of ovarioles containing a marked FSC was also increased with *UAS-Hop* expression to 75.2% compared to 62.1% in controls (*Figure 3H*). When the increase in EdU index was suppressed by co-expressing *UAS-Dap* with *UAS-Hop*, the increase in FSC numbers was greatly reduced (*Figure 3H*).

FSC clones expressing *UAS-CycE* had a similar increase in the average EdU index to those expressing *UAS-Hop* (40.4% vs 43.9%) (*Figure 2A,K*) and, again, FSC numbers were increased. However, the increase was more modest for CycE overexpression, with an average of 5.6 marked FSCs per ovariole by 12d and 64.3% of ovarioles containing a marked FSC (*Figure 3H*). Neither *UAS-CycE* nor *UAS-Hop* significantly altered steady-state FSC AP location (*Figure 3A*), and while *UAS-Hop* promoted conversion of FSCs to FCs (*Figure 3B*), *UAS-CycE* did not (data not shown). The larger impact of increased JAK-STAT pathway activity on FSC numbers is plausibly because increased division was promoted preferentially in anterior FSC layers (*Figure 2A,K*), which normally do not divide as frequently and are lost directly to differentiation at a lower frequency than posterior FSCs, or because the domain of dividing cells has expanded into the EC region. It is also possible that the EdU index does not reflect division rates accurately and that the FSC division rates in response to excess CycE or excess Hop are not as similar as suggested by EdU incorporation.

When STAT activity was eliminated in FSC clones, there was an average of 0.3 *stat* FSCs per germarium and 25.3% of germaria retained a marked FSC after 12d (*Figure 3H*). These are large deficits compared to controls (3.2 FSCs, 62.1% of ovarioles), and similar to *cycE* partial loss of function mutants (0.5 FSCs, 29% of ovarioles), which also drastically reduce FSC proliferation. When we tested *kibra stat, wts stat,* and *UAS-CycE stat* genotypes, the average number of FSCs increased to 2.1 per germarium with 68.5% of germaria containing an FSC clone. An improved persistence of FSCs was expected but the magnitude of rescue was surprisingly large if considering only FSC division rates. This disparity was even more pronounced when examining FSC competition for *kibra stat* mutants expressing *UAS-CycE*, which had an average EdU index about 64% of wild-type. Here, the average number of marked FSCs was extremely high at 15.9 per germarium and 94% of ovarioles included marked FSCs (*Figure 3E,H*). The remarkable persistence and amplification of these FSCs shows that factors other than division rate are also major determinants of FSC competition. Specifically, the dramatically increased competitive success of FSCs lacking STAT activity for a given division rate very likely results from reduced conversion to FCs. Reduced conversion to FCs results from both the infrequent presence of FSCs in layer 1 (*Figure 3A*) and the markedly lower conversion of layer 1 FSCs into FCs when FSCs lack STAT activity (*Figure 3C*). The virtual sealing off of this conduit, which is normally the major route for FSC loss, allows marked FSCs to accumulate despite dividing at rates lower than their normal unmarked FSC neighbors.

Although the survival and amplification of FSCs lacking STAT activity were increased towards, and then beyond normal by relatively modest restoration of division rates, those FSCs still had much reduced physiological activity, measured by continued production of FCs, with very few ovarioles containing both FSCs and FCs (3.5% for *stat*, 9.5% for *stat* with *kibra*, *wts* or *UAS-CycE*, 17.0% for *kibra UAS-CycE stat*, compared to 42.8% for controls) (*Figure 3D,E,H*). Thus, interfering with the normal coordination of FSC division and conversion to FCs in response to JAK-STAT pathway activity by adding genetic modifiers of division alone led to extensive amplification of unproductive FSCs. In many of these ovarioles, egg chambers were surrounded entirely by unmarked cells and had an abnormal, elongated morphology (*Figure 3E*), perhaps suggestive of a deficiency in overall FC production.

When excess Fas3 was expressed in *kibra UAS-CycE stat* FSCs, doubling conversion of layer 1 FSCs to FCs and restoring rates of FC production towards normal values (*Figure 3B*), the average number of FSCs per germarium declined sharply from 15.9 to 2.4 (*Figure 3H*). At the same time, the percentage of ovarioles with at least one FSC at 12d declined from 94% to 59% but the proportion of ovarioles with FSCs and FCs increased from 17% to 31% (*Figure 3H*). The response to excess Fas3 provides further evidence of the large impact of the rate of FC production on both FSC numbers and the ability of FSCs to fulfill their physiological role. It also demonstrates that appropriate magnitudes of artificial stimulation of both FSC division and FSC differentiation to FCs can partially substitute for the normal coordination of these rates by JAK-STAT signaling to bring about roughly normal FSC behavior.

## Wnt signaling primarily influences FSC position and differentiation

The role of Wnt signaling in regulating FSC behavior has already been examined in the context of a revised model of FSC numbers, locations and properties. A *Fz3-RFP* reporter demonstrated that Wnt pathway activity decreases in strength across the FSC region, in the anterior to posterior direction (*Reilein et al., 2017*; *Wang and Page-McCaw, 2014*). FSCs with a null *arrow* (*arr*) mutation to eliminate the Wnt pathway response and *axin* (*axn*) or *Adenomatous Polyposis Coli* (*apc*) mutations to constitutively activate the Wnt pathway in MARCM clones (*Reilein et al., 2017*), all illustrated a strong effect of higher Wnt signaling activity favoring anterior FSC locations and greater conversion to ECs; 77.6% of *arr* FSCs but only 15–20% of *axn* and *apc* FSCs were observed in layer 1, while 9.1 *axn* and *apc* ECs and 0.1 *arr* ECs were observed per germarium, compared to an average of 1.5 ECs for controls (*Reilein et al., 2017*).

Here we also tested the effect of reducing rather than eliminating Wnt pathway activity by expression of a *UAS-dnTCF* transgene (*van de Wetering et al., 2002*) in clones. We found that 63.0% of *UAS-dnTCF* FSCs were observed in layer 1 (compared to 48.8% for controls), a significant change but less pronounced than for *arr* FSCs, which showed a layer 1 occupancy of 79.3% across three replicates (n = 237 cells), including two additional tests not previously reported (*Figure 4A*). We also tested an additional *axn* replicate and confirmed layer one occupancy to be greatly decreased, to 20.6% at 12d. Occupancy of layer one was slightly higher (31.0%) for *axn* FSCs that also expressed excess CycE (*Figure 4A*) to increase division rates (from 7.4% to 15.3% towards control 25.0% EdU frequency (*Figure 4E*)), consistent with the evidence presented earlier of low division rates favoring more anterior locations.

We also used the 'immediate FC' method to measure FC production. We found that *arr* layer 1 FSC clones had a significantly elevated probability (76.9% compared to 61.6% in controls) of becoming an FC, (*Figure 4B*). By contrast, reducing Wnt pathway activity with *UAS-dnTCF* did not increase conversion of layer 1 FSCs to FCs (52.9% probability). We also observed that FSCs with increased Wnt activity showed only a 22.4% likelihood of becoming an FC, a roughly threefold decrease from control values (*Figure 4B*). We therefore extend previous conclusions to surmise that the AP location of FSCs in a competitive environment of normal FSCs is altered by reduction, elimination or increases of Wnt pathway activity. By measuring the conversion of layer 1 FSCs to FCs as an independent parameter for the first time, we also found that increased Wnt pathway activity strongly reduced FC production from posterior FSCs and that only the most severe reductions in Wnt pathway activity enhanced FC production. These results suggest that the magnitude of Wnt pathway activity affects AP migration over the whole FSC domain, where Wnt signaling is graded, and that the decline of pathway activity to near zero values at the posterior margin of the FSC domain is a significant determinant of the FSC to FC transition.

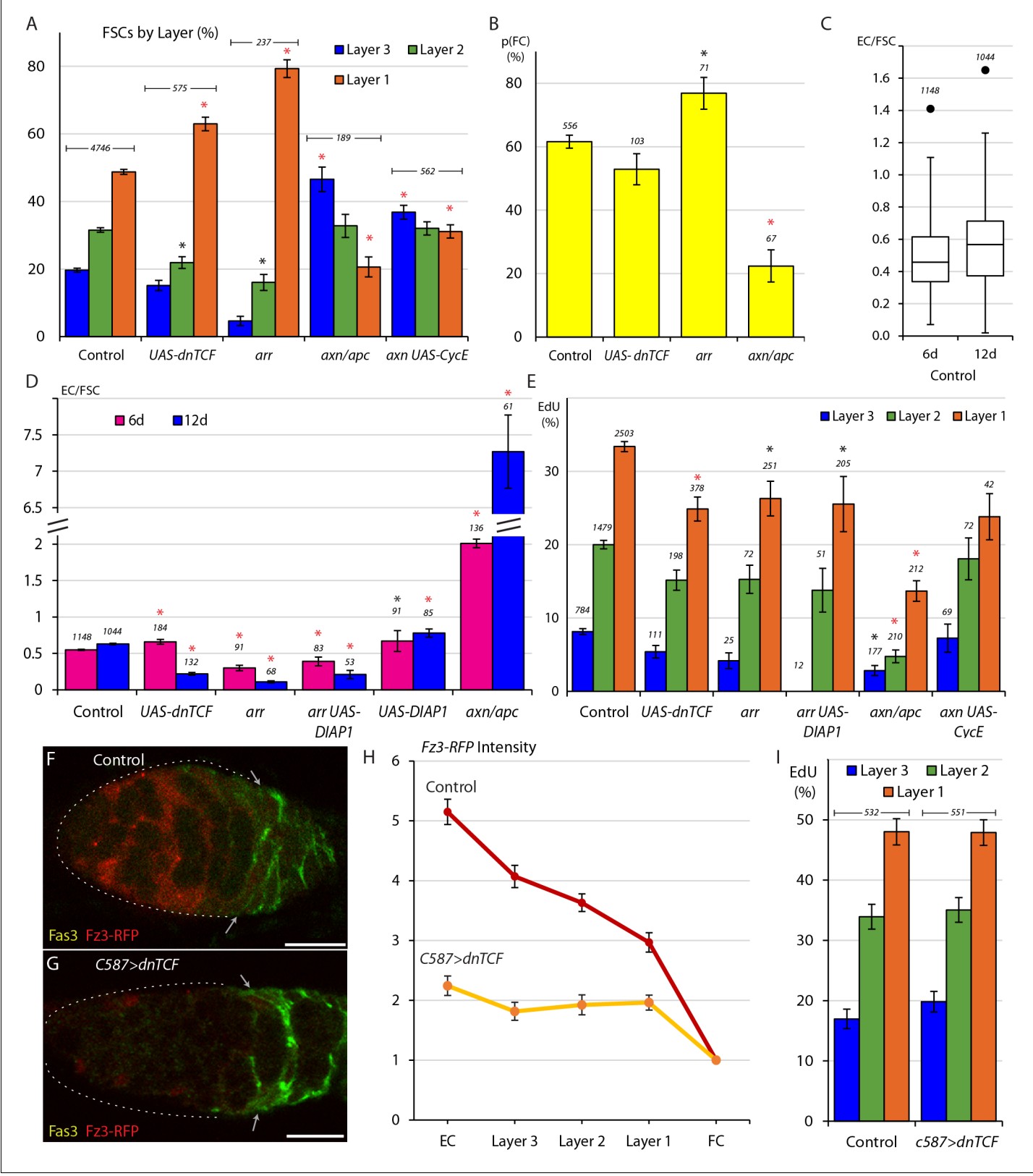

**Figure 4.** Wnt signaling opposes FC production and promotes anterior FSC location and EC production. (**A**) Relative frequency of marked FSCs in each layer for the indicated genotypes of MARCM lineages, with the number of total FSCs scored for each genotype and significant differences to control values (black asterisks, p<0.05, red asterisks, p<0.001). (**B**) Average probability of a layer 1 FSC becoming an FC during a single budding cycle for the

*Figure 4 continued on next page*

*Figure 4 continued*

indicated MARCM lineage genotypes with the number of informative germaria scored and significant differences to control values (black asterisks, p<0.05, red asterisks, p<0.001). (C) Box-and-whisker plot (median, first and third quartile, minimum and maximum) of ECs produced per anterior FSC from 0-6d and 0-12d across all MARCM controls (n = 31 experiments) with a single outlier and the number of germaria scored. (D) ECs produced per anterior FSC from 0-6d (magenta) and 0-12d (blue) for the indicated MARCM lineage genotypes with the total number of informative germaria scored and significant differences to control values (black asterisks, p<0.05, red asterisks, p<0.001). (E) EdU incorporation frequency into FSCs of layers 1–3 and ECs for the indicated genotypes of MARCM lineages with number of cells scored above each column and significant differences between FSC layers indicated (black asterisks, p<0.05, red asterisks, p<0.001). The statistical significance of the EdU index of each FSC layer as a fraction of overall EdU index relative to controls was also calculated for all genotypes. No significant differences were found, indicating a gradient of EdU incorporation that is not significantly different from controls. (F–G) Fz3-RFP reporter of Wnt pathway activity (red) (F) normally declines in strength from anterior to posterior but (G) was mostly eliminated after 3d of *UAS-dnTCF* expression with *C587-GAL4*. (H) Average Fz3-RFP intensity for control (n = 23 germaria) and C587>dnTCF (n = 22 germaria) genotypes. (I) EdU incorporation frequency into FSCs in layers 1–3 for the indicated genotypes (number of DAPI-labeled nuclei scored is above each column). The statistical significance of the EdU index of each FSC layer as a fraction of overall EdU index relative to controls was also calculated for *C587>dnTCF*. No significant differences were found, indicating a gradient of EdU incorporation that is not significantly different from controls. All scale bars are 10 µm.

The online version of this article includes the following source data for figure 4:

**Source data 1.** Numerical data for graphs in *Figure 4*.

## ECs derived from FSCs turn over relatively rapidly

To evaluate EC production from anterior FSCs, we calculated the average ratio of marked ECs per marked anterior (layer 2 or 3) FSC. We calculated this ratio for all germaria that retained at least one marked FSC, so that there was a possibility of EC production throughout the period scored. The average number of marked anterior FSCs (aFSCs) for control clones was slightly higher at 12d (2.7) than at 6d (2.2), as expected because more ovarioles lack any FSCs at 12d and are not included (*Figure 1G*). We took the number of marked anterior FSCs per germarium at 0d to be equal to those scored at 6d because almost all germaria have FSCs at 6d. We then estimated the average number of anterior FSCs present during the 0-12d period as the average of the number present at 0d (and measured at 6d) and the number measured at 12d.

In controls, we found that the EC/aFSC ratio was 0.55 (SE 0.30; SEM 0.01) for the period from 0-6d and 0.63 (SE 0.33; SEM 0.01) for the period from 0-12d (*Figure 4D*; median and other statistical measures are in *Figure 4C*). If ECs were produced by FSCs at a constant rate and all labeled ECs accumulated without loss, we would expect the 0-12d ratio to be double the 0-6d ratio. The observed percentage increase was much lower (15%) across 31 control tests, suggesting that marked ECs must also be lost at a significant frequency. The same inference is apparent from looking at the number of marked ECs per germarium at 6d (1.2) and at 12d (1.5) without any correction for the slightly greater number of anterior FSCs in germaria that retain FSCs at 12d (but still considering the same set of samples with at least one FSC). It appears that by 12d the number of marked ECs is at, or approaching a steady-state where the rate of production is matched by the rate of loss. This occurs when the average number of anterior FSCs (2.7) is almost double the number of marked ECs (1.5), suggesting that the rate of loss of marked ECs is greater than the rate of conversion from anterior FSCs on a per cell basis. Clearly, if this rate of EC loss applied to the whole population of over 30 ECs, far outnumbering the total of about eight anterior FSCs, there would be a severe net loss of ECs over time. We therefore deduce that the relatively high turnover that we infer for marked ECs must apply only to ECs newly-produced from FSCs and not to the bulk EC population present at the time of adult eclosion. It is certainly plausible that an FSC that moves into EC territory might indeed often return to its former FSC position or be unable to survive for long in EC territory if it does not associate with a germline cyst to receive key survival signals (*Kirilly et al., 2011*).

Apoptosis is observed in normal germaria at a low frequency (*Pritchett et al., 2009*) with the fraction of ECs undergoing apoptosis at any instant reported as 1.5% (*Wang and Page-McCaw, 2018*) or about 0.5% (19% of germaria, each with about 35 ECs) (*Kirilly et al., 2011*) by TUNEL labeling. The majority of apoptosis is observed in the neighborhood of the EC/FSC boundary. We sought to test whether the turnover of marked ECs was due to apoptosis by expressing the inhibitor of apoptosis, DIAP1 in otherwise wild-type clones. If EC turnover were reduced we might expect to see a greater number of ECs accumulating at all time points, including continued significant accumulation beyond 6d. We observed a small increase in marked EC numbers, whether measured per

germarium at 6d (1.5) or 12d (1.7), or per anterior FSC at 6d (0.67) or 12d (0.78) (*Figure 4D*). These results suggest that apoptosis does contribute to the turnover of marked ECs but it does not appear to be the major factor, with EC production over 12d far short of doubling EC production over the first 6d. Other forms of cell death may play a role but it is perhaps most likely that an FSC that moves into the EC region frequently returns to the FSC domain.

## Net EC production is promoted by Wnt and opposed by JAK-STAT pathway activities

We measured EC production from FSCs of various altered genotypes as described for controls, using the number of anterior FSCs at 6d for controls in each experiment as the best estimate of the number of anterior FSCs at 0d for all genotypes, so that the average number of anterior FSCs present during 0-6d and 0-12d could be calculated. We found that EC production was drastically reduced for *arr* mutant FSCs at 12d (0.11 ECs per anterior FSC compared to 0.63 for controls) and less severely at 6d (0.30 vs 0.55) (*Figure 4D*). The relatively high yield of ECs at 6d is likely because several ECs were produced in the first day or two before wild-type Arr protein was depleted and cell behavior was altered. Similarly, in FSCs expressing *UAS-dnTCF* there was a severe loss of marked ECs at 12d relative to controls (0.22 vs 0.63) but not at 6d (*Figure 4D*); here some time is likely required to accumulate sufficient dnTCF protein to inhibit Wnt pathway activity substantially. These data indicated that reduction, and especially loss of Wnt signaling greatly reduced the net conversion of marked anterior FSCs to ECs. This may be due to reduced conversion of anterior FSCs to ECs, increased turnover of ECs derived from FSCs, or both.

Loss of Wnt pathway activity is known to elicit apoptosis in ECs (*Wang et al., 2015*; *Wang and Page-McCaw, 2018*). We found that when we expressed *UAS-DIAP1* in *arr* mutant FSCs, EC accumulation was increased relative to *arr* alone at both 6d (0.39 vs 0.30) and 12d (0.21 vs 0.11, p=0.06) but remained much below control values (*Figure 4D*). The continued deficit in EC accumulation for *arr* mutant cells expressing DIAP1 and that of cells expressing *UAS-dnTCF* suggests that the equilibrium of conversion between anterior FSCs and ECs is tilted strongly towards FSCs when Wnt pathway activity is reduced.

When Wnt pathway activity was increased using the *axn* and *apc* mutations we found the average ratio of ECs per anterior FSC to be 2.0 from 0-6d and 7.3 from 0-12d, revealing a greatly elevated rate of EC accumulation (*Figure 4D*). Again, perdurance of wild-type Axn or Apc proteins over the earliest period may account for the less dramatic rate of EC accumulation over the first 6d. The rapid addition of labeled ECs over 6-12d and the observation that FSC numbers eventually decline towards zero, leaving many labeled ECs suggest that there is little or no turnover of newly-produced ECs. Thus, any normal reversion of FSCs moving into EC locations back to FSC locations is strongly opposed by high levels of Wnt pathway activity. For all genotypes with altered Wnt pathway activity the marked cells in EC locations did not incorporate EdU and therefore do exhibit one of the key characteristics that distinguishes them as ECs rather than FSCs. We did not mark cellular processes and did not therefore ascertain whether these cells encapsulated germline cysts.

We also investigated JAK-STAT signaling and EC production dynamics. When STAT activity was eliminated, we found that EC production increased to 0.88 ECs per anterior FSC from 0-6d and 1.2 ECs per anterior FSC from 0-12d (*Figure 3B*), showing that the normal equilibrium between anterior FSCs and ECs was shifted significantly towards ECs, potentially by both increasing EC production and reducing EC loss.

The number of ECs initially produced from FSCs cannot be measured accurately for FSCs expressing *UAS-Hop* because increased JAK-STAT pathway activity sometimes induces the subsequent division of cells in the EC domain. We found that adding *UAS-Dap* to *UAS-Hop* reduced ectopic division in the EC domain from an EdU index of 12.4% to 5.9%. EC production from 0-6d was lower for *UAS-Hop UAS-Dap* FSC lineages than controls, suggesting some reduction in conversion to ECs (*Figure 3B*). The increased yield of marked cells in the EC region by 12d presumably results from division of some of those cells. These results, together with the unaltered AP distribution of FSCs with excess JAK-STAT activity and the anterior accumulation of *stat* mutant FSCs (*Figure 3A*) suggest that a certain minimal level of JAK-STAT pathway activity is required to prevent unbalanced anterior migration of FSCs and accelerated net conversion of anterior FSCs to ECs.

## Wnt signaling can reduce FSC division but does not greatly affect the magnitude or pattern of FSC division under normal conditions

It has previously been reported that loss of Wnt signaling reduced FSC division by a small amount, with most measured FSCs in layer 1, while increased Wnt pathway activity, measured mostly in anterior FSCs, greatly decreased FSC proliferation, with results normalized in each case for FSC locations (*Reilein et al., 2017*). To examine the effects of reduced signaling more comprehensively in anterior FSCs we looked at more *arr* mutant samples, including those additionally expressing DIAP1, and four independent experiments where Wnt signaling was reduced by expression of *dnTCF*, which results in a less pronounced, but still significant, posterior shift of FSCs than complete inhibition of Wnt signaling (*Figure 4A*). We found that EdU incorporation was slightly reduced in all layers for FSCs lacking *arr* activity (26.3%, 15.3%, 4.0% for layers 1–3) or expressing *dnTCF* (24.9%, 15.2%, 5.4% for layers 1–3) relative to controls (33.4%, 20.0%, 8.2% for layers 1–3); the differences in layer one were statistically significant (*Figure 4E*). In all experiments where Wnt signaling was reduced or eliminated there was a clear posterior to anterior gradient of EdU labeling, as in normal FSCs with no statistically significant difference from controls in the relative EdU index of any one layer relative to the whole FSC population (*Figure 4E*).

In response to increased Wnt pathway activity, the EdU index was substantially reduced in all layers (13.7%, 4.8%, 2.8% for layers 1–3) but a posterior to anterior gradient was still evident (*Figure 4E*). To test the effect on graded proliferation further we combined loss of *axn* with *UAS-CycE* in an attempt to restore overall FSC proliferation towards wild-type levels. Excess CycE indeed doubled EdU labeling frequency overall (from 7.3% to 15.3%; control was 25%) and a robust posterior to anterior gradient was still evident (23.8%, 18.1%, 7.3% for layers 1–3) with no statistically significant difference from controls in the relative EdU index of any one layer relative to the whole FSC population for *axn/apc* or *axn UAS-CycE* genotypes (*Figure 4E*).

Finally, to assess the contribution of graded Wnt signaling to the A/P pattern of graded proliferation, we reduced Wnt signaling globally by expressing the *UAS-dnTCF* transgene with the *C587-GAL4* driver (*C587>dnTCF*). As both normal Wnt pathway activity and *C587-GAL4* expression decline from anterior to posterior across the FSC domain, this manipulation ought in theory to flatten or eliminate the normal Wnt gradient to produce a roughly even, low level of pathway activity. Fz3-RFP reporter expression showed that Wnt signaling was considerably reduced and was close to uniform across the three FSC layers (*Figure 4F–H*). Under these conditions there was very little difference in EdU incorporation in any FSC layer when compared to controls (*Figure 4I*). This result is consistent with the evidence from MARCM clones that graded FSC proliferation does not rely on graded Wnt signaling.

## JAK-STAT and Wnt pathways do not directly affect one another cell autonomously

As both the JAK-STAT and Wnt signaling pathways play important roles in the regulation of FSCs, we asked whether regulation by each pathway is accomplished independently. We measured whether genetic manipulation of the Wnt pathway influenced JAK-STAT pathway activity cell autonomously by inducing GFP-positive MARCM clones that also expressed a STAT-eRFP reporter, which has STAT-responsive promoter sequences from the *Socs36E* gene (*He et al., 2019*). Reciprocal tests measured effects of JAK-STAT pathway alterations on Fz3-RFP reporter activity. In these tests, we measured the signal intensity of the reporters in marked cells relative to unmarked neighbors (*Figure 5—figure supplements 1* and *2*). Cells were considered neighbors if they would have been scored in the same FSC layer, and if they were captured within the same z-section during confocal imaging. Samples were examined 6d after clone induction so that all genotypes included marked cells in a full range of FSC locations.

For genotypes affecting JAK-STAT signaling components, STAT-RFP intensity was significantly altered when compared to neighbors, as expected. Reporter activity was significantly lower for *stat* mutant cells (42.4%, p=0.001) and was not altered in control clones (*Figure 5—figure supplement 1A,B*). Cells expressing *UAS-Hop* had STAT-RFP expression roughly twice that of normal neighbors. The *stat* genotype used is expected to prevent all stimulated JAK-STAT pathway activity. The measured residual RFP likely corresponds to a combination of basal reporter expression that is not dependent on JAK-STAT pathway activity and any perduring RFP that was induced before normal

STAT protein and pathway activity were fully depleted during clone expansion. Taking RFP levels in *stat* mutant clones as the effective null condition, the change in STAT-RFP levels in *UAS-Hop* clones corresponds to an increase of more than two-fold in normal JAK-STAT pathway activity (roughly 160%/60% = 2.7). Neither FSCs expressing *UAS-dnTCF* nor those with *axn* mutations (together with *UAS-CycE* in order to increase FSC numbers) had significantly altered STAT-RFP expression suggesting that the magnitude of Wnt pathway activity does not cell autonomously affect JAK-STAT pathway signal transduction (*Figure 5—figure supplement 1A,B*).

For genotypes affecting Wnt signaling components, Fz3-RFP intensity was significantly altered when compared to neighbors, as expected. Reporter activity was significantly lower for *arr* mutant cells (39.4%, p=0.001) and for cells expressing *UAS-dnTCF* (63.1%, p=0.01) (Fig. S2). The null *arr* genotype used is expected to prevent all stimulated Wnt pathway activity (*Wehrli et al., 2000*), so residual RFP likely corresponds to perduring RFP and basal reporter activity. If RFP levels in *arr* mutant clones are taken as the effective null condition, *UAS-dnTCF* expression reduced Wnt pathway activity to below 40% of normal levels (23.4% reduction in a range of 60.3% from null to normal) in MARCM FSC lineages and the measured elevation of pathway-induced Fz3-RFP levels in *axn* clones (169.4%, p=0.02) corresponds to a roughly two-fold increase in normal Wnt pathway activity (129.5% change compared to a normal range of 60.3%)(*Figure 5—figure supplement 2A*). Neither decreased (*stat* and *kibra UAS-CycE stat*) nor increased (*UAS-Hop*) JAK-STAT pathway activity significantly altered Fz3-RFP expression, suggesting no direct, cell autonomous effect of JAK-STAT signaling on Wnt signal transduction (*Figure 5—figure supplement 2A,B*).

We then asked how manipulating both pathways within the same FSC would influence FSC behavior by measuring proliferation, position, and differentiation when both JAK-STAT and Wnt signaling activity were altered in FSC clones.

## FSC division rates; inhibition by Wnt is suppressed by JAK-STAT pathway activity

Reduction or elimination of Wnt signaling alone caused a small reduction in EdU incorporation in MARCM clones (*Figure 4E*). It was therefore surprising that expressing *UAS-dnTCF* increased EdU incorporation in FSCs lacking STAT activity (from 2.4% to 4.5%). This effect was more striking in STAT-deficient genotypes with higher division rates, elevating the EdU index of FSCs with *stat* mutations together with *kibra, wts,* or *UAS-CycE* from an average of 4.9% to 16.2% (*Figure 5A*), close to the value observed for otherwise normal FSCs expressing dn-TCF (19.0%). In other words, a five-fold reduction of EdU incorporation (5% vs 25%) due to the *stat kibra/wts/UAS-CycE* genotype was largely suppressed when Wnt pathway was reduced by *UAS-dnTCF*.

By contrast, the increased FSC division induced by increased JAK-STAT pathway activity (43.9% EdU index) was not significantly altered by addition of dnTCF (46.3%) and was only slightly reduced by loss of *arr* function (37.7%) (*Figure 5B*; *Figure 5—figure supplement 3A*; *Figure 6*). Ectopic EdU labeling in the EC region was also largely unaltered by reducing Wnt pathway activity (increasing to 16.7% from 12.4%) (*Figure 5B*). EdU labeling of FSCs with increased JAK-STAT pathway activity was also only slightly reduced by genetically increasing Wnt pathway activity (from an EdU index of 43.9 to 38.3% for *axn UAS-Hop*) to a level far above that observed for *axn/apc* mutant FSCs (7.3%) (*Figure 5B*; *Figure 5—figure supplement 3B,C*). Even at a lower temperature of 22C, where GAL4 and consequently *UAS-Hop* activities are lower, the EdU index for *axn UAS-Hop* FSCs was higher than control values (28.2% vs. 18.9%) and not much lower than for *UAS-Hop* alone (35.1%) (*Figure 5—figure supplement 3D*). At both temperatures the EdU index of marked cells in the EC region was higher for *axn UAS-Hop* than for *UAS-Hop* lineages (*Figure 5B*; *Figure 5—figure supplement 3D*).

These results indicate that normal Wnt pathway activity inhibits FSC division under artificial conditions of removing STAT activity, even in locations (layer 1) where Wnt pathway activity is quite low (*Figure 5A*). Under otherwise normal conditions, genetically increasing Wnt pathway activity to a level that approximates or slightly exceeds the highest physiological levels observed in the germarium, strongly inhibited FSC division but this inhibition was robustly overridden by genetically increasing JAK-STAT pathway activity (*Figure 5B*). Thus, inhibitory actions of the Wnt pathway can be strong and dose-dependent but are strongly suppressed by JAK-STAT pathway activity under conditions when both pathways are unaltered or both are artificially elevated.

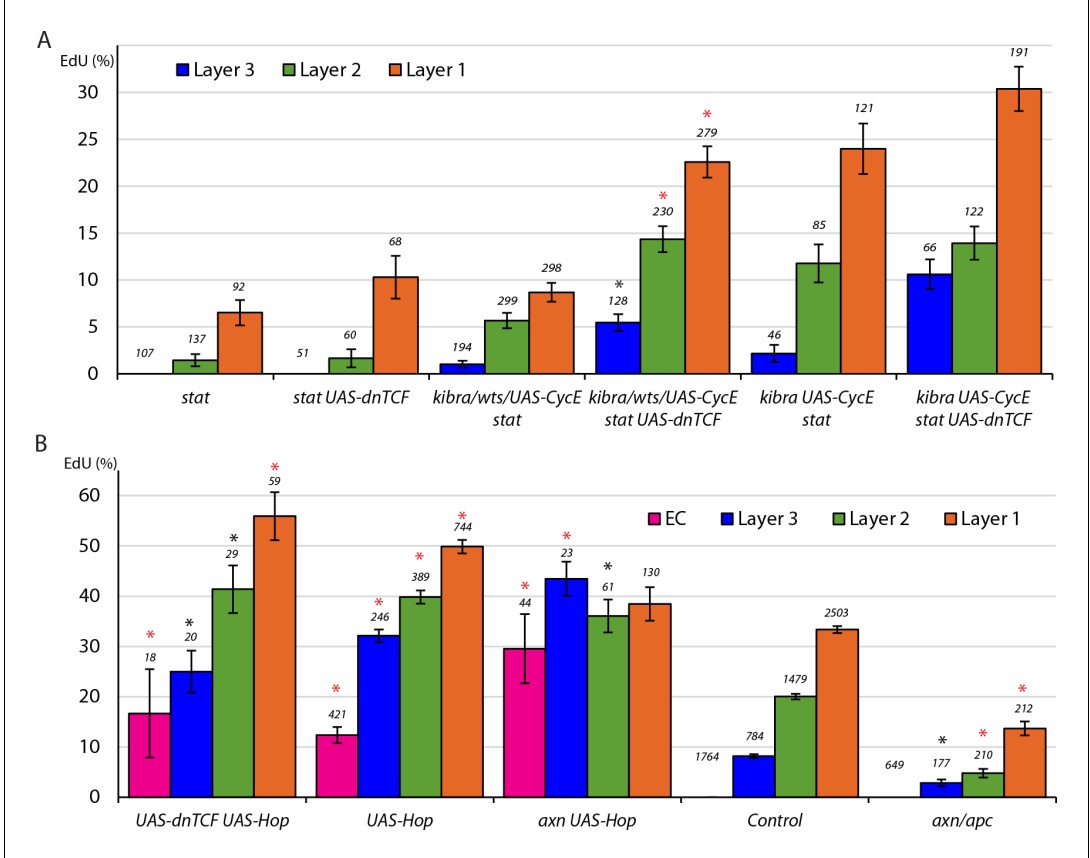

**Figure 5.** Wnt pathway activity reduces FSC division only in the absence of JAK-STAT pathway activity and JAK-STAT overrides inhibition by the Wnt pathway when both are in excess. (A, B) EdU incorporation frequency into FSCs of layers 1–3 and ECs for the indicated genotypes of MARCM lineages with number of cells scored above each column. (A) Significant differences between genotypes with and without *UAS-dnTCF* (black asterisks, p<0.05, red asterisks, p<0.001) are indicated for individual FSC layers. The statistical significance of the EdU index of each FSC layer as a fraction of overall EdU index relative to controls was also calculated for all genotypes. No significant differences were found, indicating a gradient of EdU incorporation that is not significantly different from controls. (B) Significant differences from control values are indicated for each FSC layer (black asterisks, p<0.05, red asterisks, p<0.001). See also *Figure 5—figure supplements 1–3*.

The online version of this article includes the following source data and figure supplement(s) for figure 5:

**Source data 1.** Numerical data for graphs in *Figure 5*.

**Figure supplement 1.** STAT-RFP intensity is not affected by Wnt pathway activity.

**Figure supplement 1—source data 1.** Numerical data for graphs in *Figure 5—figure supplement 1*.

**Figure supplement 2.** Fz3-RFP intensity is not affected by JAK-STAT pathway activity.

**Figure supplement 2—source data 1.** Numerical data for graphs in *Figure 5—figure supplement 2*.

**Figure supplement 3.** Increased JAK-STAT signaling promotes FSC division even when Wnt pathway activity is altered.

**Figure supplement 3—source data 1.** Numerical data for graphs in *Figure 5—figure supplement 3*.

In all samples where FSCs lacked STAT activity and expressed dnTCF there was a robust posterior to anterior gradient of EdU labeling (*Figure 5A*). This included genotypes with overall division rates close to controls (*kibra UAS-CycE stat* plus *UAS-dnTCF*; 21.6% vs 25% control EdU index). Thus, provided overall FSC division is bolstered by excess CycE and Yki activation, there remains a roughly normal FSC proliferation gradient in the complete absence of one major graded signaling pathway (JAK-STAT) and reduction of another (Wnt).

## Potent acceleration of FC production from increased JAK-STAT combined with absent Wnt pathway activity

Increased JAK-STAT activity favored conversion of layer 1 FSCs to FCs (78.1% probability) and this was further increased by loss of *arr* activity (90.3% for *arr UAS-Hop* FSCs) to a level higher than

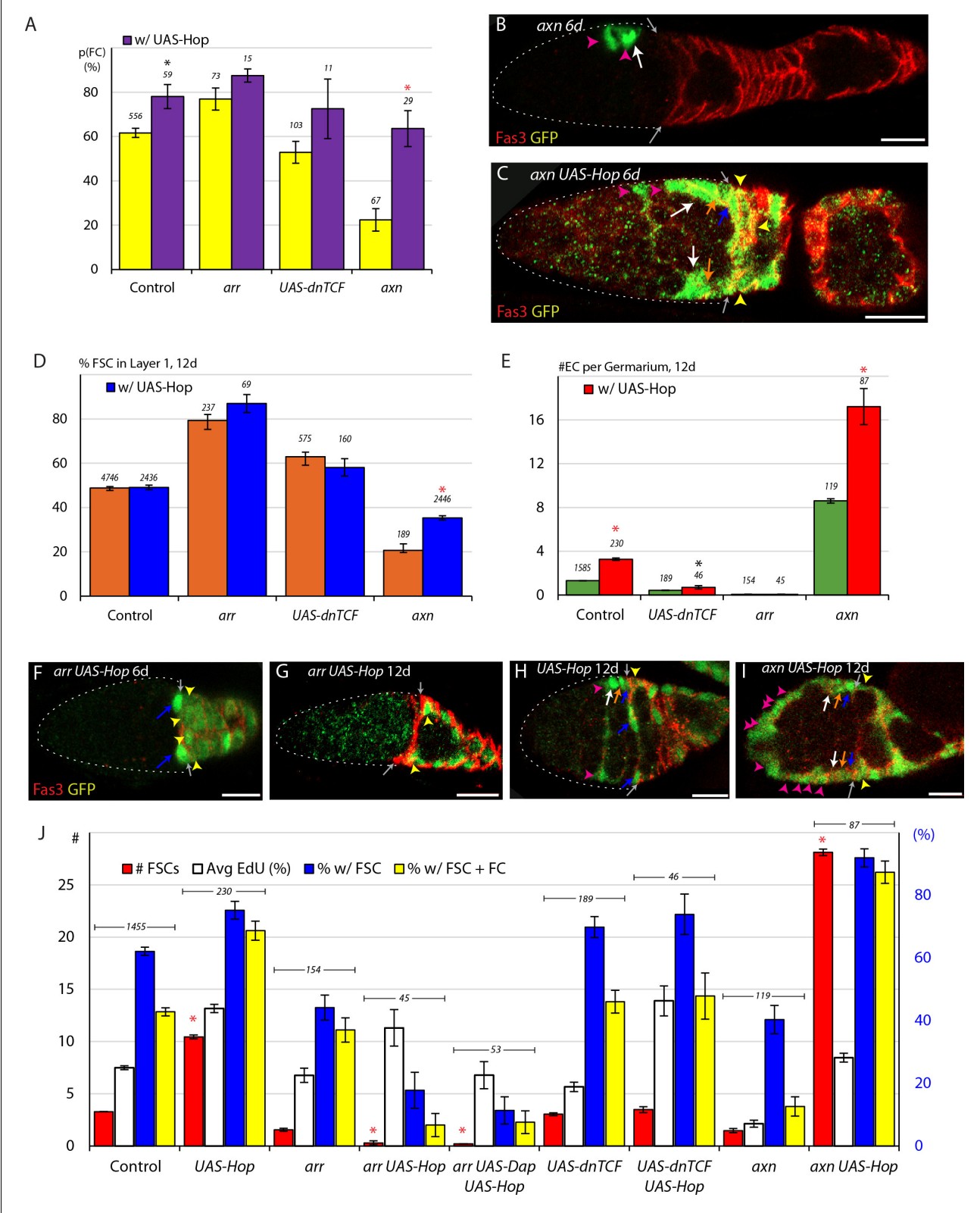

**Figure 6.** Promotion of FC production by the JAK-STAT pathway overrides the opposing influence of increased Wnt pathway activity and synergizes with loss of Wnt pathway activity to cause dramatic loss of highly proliferative FSCs. (**A**) Average probability of a layer 1 FSC becoming an FC during a single budding cycle for the indicated MARCM lineage genotypes with the number of informative germaria scored and significant differences resulting from the presence (purple) of *UAS-Hop* (black asterisks, p<0.05, red asterisks, p<0.001). (**B, C**) *axn* mutant MARCM lineages (green) at 6d, with the Fas3

*Figure 6 continued on next page*

*Figure 6 continued*

(red) anterior border (gray arrows indicated) generally include, as here, anterior FSCs (layer 3, white arrows), and ECs (magenta arrowheads) but (C) addition of *UAS-Hop* resulted in more marked FSCs, including layer 1 (blue arrows) and layer 2 (orange arrows) FSCs, and marked FCs (yellow arrowheads). (D) Proportion of FSCs in layer one for the indicated MARCM lineage genotypes, with the number of FSCs scored and significant differences resulting from the presence (blue) of *UAS-Hop* (red asterisks, p<0.001). (E) Number of ECs per germarium for the indicated MARCM lineage genotypes at 12d, with the number of germaria scored and significant differences resulting from the presence (red) of *UAS-Hop* (black asterisks, p<0.05, red asterisks, p<0.001). (F–I) MARCM lineages (green) with the Fas3 (red) anterior border indicated (gray arrows), commonly showed, as here, (F) only layer 1 FSCs (blue arrows) and FCs (immediate FCs, yellow arrowheads) for *arr UAS-Hop* at 6d and (G) loss of FSCs, leaving only labeled FCs (yellow arrowheads) by 12d and (H, I) a large increase of labeled cells in EC locations (magenta arrowheads) when Wnt pathway activity is increased (*axn*) on top of increased JAK-STAT pathway (*UAS-Hop*), supplementing the many labeled FSCs in layers 1 (blue arrows), 2 (orange arrows) and 3 (white arrows) and FCs (yellow arrowheads). (J) Number of FSCs per germarium (red) using y-axis scale on the left, percentage of FSCs incorporating EdU (aggregating all layers, white), percentage of ovarioles with a marked FSC (blue) and percentage of ovarioles with a marked FSC and marked FCs (yellow) (percentage y-axis scale in blue on the right) for the indicated genotypes, with the number of germaria scored at 12d (EdU was scored at 6d) and significant differences for the number of FSCs (red asterisks, p<0.001) compared between genotypes with and without *UAS-Hop*. All scale bars are 10 μm. See also *Figure 6—figure supplements 1–2*.

The online version of this article includes the following source data and figure supplement(s) for figure 6:

**Source data 1.** Numerical data for graphs in *Figure 6*.
**Figure supplement 1.** Decreasing Wnt signaling increases FSC division and posterior location but does not strongly alter loss of FC production or increased FC production in the absence of JAK-STAT signaling.
**Figure supplement 1—source data 1.** Numerical data for graphs in *Figure 6—figure supplement 1*.
**Figure supplement 2.** Excess JAK-STAT pathway activity promotes conversion of layer 1 FSCs to FCs even when Wnt pathway activity is increased.
**Figure supplement 2—source data 1.** Numerical data for graphs in *Figure 6—figure supplement 2*.

induced by Wnt signaling deficiency alone (76.9% for *arr* FSCs) (*Figure 6A*). Thus, the combination of eliminating the normally low levels of Wnt signaling and increasing the already high levels of JAK-STAT pathway in layer 1 FSCs potently accelerated and almost mandated conversion of layer 1 FSCs to FCs. Moreover, despite accelerated loss from layer one to become FCs, 87.1% of *arr UAS-Hop* FSCs were found in this layer, representing an enhancement of the bias seen for *arr* FSCs (79.3%) (*Figure 6D*), and further accelerating overall FC production. Expression of dnTCF did not increase the probability of layer 1 FSC to FC conversion in the presence of either increased JAK-STAT pathway activity (*Figure 6A*) or in FSCs lacking STAT activity (*Figure 6—figure supplement 1A*). Artificially increasing Wnt pathway activity in FSCs strongly inhibited conversion to FCs (21.5% probability), but this inhibition was entirely negated by increasing JAK-STAT pathway activity in *axn UAS-Hop* FSCs, whether tested at 25°C (*Figure 6A*) or 22°C (*Figure 6—figure supplement 2*). Thus, reducing Wnt pathway activity to zero synergized with elevated JAK-STAT pathway activity to potently drive FSCs to become FCs, while smaller decreases achieved with dnTCF were without major consequence. High JAK-STAT activity also overcame the normally strong inhibition of FC production by elevated Wnt pathway activity.

EC production was reduced for *arr* FSCs, increased dramatically for *axn* FSCs and increased less prominently for *stat* FSCs. Reduction of Wnt signaling with dnTCF reduced EC production alone by 12d (*Figure 4D*) but did not diminish the increased EC production of FSCs lacking STAT activity (*Figure 6—figure supplement 1E*). The effects of *UAS-Hop* on EC production per anterior FSC cannot be quantified accurately because some of these marked cells in the EC region divide. Nevertheless, measurement of the total number of marked ECs provides some guidance. The average number of marked cells in the EC region per germarium at 12d, which was 1.3 for controls and 3.3 for *UAS-Hop* lineages, was greatly reduced for *arr UAS-Hop* (0.07), reduced to a lesser degree for *UAS-dnTCF UAS-Hop*, and greatly increased for *axn UAS-Hop* (17.2) FSC lineages (*Figure 6E,G–I*), showing that EC production is still highly responsive to changes in Wnt pathway activity in both directions even when JAK-STAT pathway activity is elevated. Thus, epistasis tests reveal a primary role of Wnt signaling for differentiation decisions in anterior regions and a primary role for JAK-STAT signaling for differentiation decisions in posterior regions.

## Compound pathway perturbations reveal potent effects of both FSC division rate and flux toward FCs on FSC competition

The relatively high number and persistence of FSCs lacking STAT activity despite low division rates has been noted earlier and attributed to diminished conversion to FCs. The addition of dnTCF increased EdU labeling without increasing conversion of layer 1 FSCs to FCs and might therefore be expected to increase FSC persistence. However, for *stat* alone or together with *kibra*, *wts* or *UAS-CycE* the average number of marked FSCs at 12d was not greatly altered by addition of dnTCF (0.48 vs 0.33 for *stat* alone, 2.0 vs 2.1 for others) (*Figure 6—figure supplement 1F*). The location of FSCs did, however, shift towards layer 1 (23.5% vs 3.5% for *stat* alone, 36.9% vs 29.7% for others) (*Figure 6—figure supplement 1B–D*) and this would be expected to increase FC production even for a constant rate of conversion of layer 1 FSCs to FCs. Thus, increased division of *stat* mutant FSCs in response to dnTCF plausibly did not increase FSC numbers because increased division was offset by a modest increase in FC production. FSCs with *kibra stat* and *UAS-CycE* were already present at saturating normal numbers (15.9) by 12d and that number was not significantly altered by the presence of *UAS-dnTCF* (16.9) (*Figure 6—figure supplement 1F*).

Elevated JAK-STAT activity increased the average FSC number to 10.4 per germarium, largely by increasing division rates (*Figure 6J*). EdU incorporation was only slightly reduced for *arr UAS-Hop* FSCs and unchanged for *UAS-dnTCF UAS-Hop* FSCs. However, complete Wnt pathway inhibition greatly increased FSC concentration in layer 1 (*Figure 6D*) and enhanced conversion from layer one to FCs (*Figure 6A,F*), with the net effect of drastically reducing the 12d average FSC population to 0.3 per germarium, with only 17.8% of ovarioles retaining any marked FSCs (*Figure 6G,J*). Wnt pathway reduction with dnTCF only modestly increased posterior accumulation of FSCs without accelerating FC production from posterior FSCs (*Figure 6A,D*) and resulted in 3.2 FSCs per germarium (*Figure 6J*). Thus, even a greatly elevated FSC division rate cannot maintain a sufficient supply of marked FSCs when they are drained by posterior flux and conversion to FCs at the high rates promoted by a combination of high JAK-STAT pathway activity and elimination of Wnt pathway activity.

With elevated Wnt signaling in *UAS-Hop* mutant lineages, FSC division rates remained abnormally high (*UAS-Hop* was largely epistatic to *axn*) (*Figure 5B*), FC production from layer 1 FSCs was roughly normal (*Figure 6A*) but FSCs were predominantly in anterior locations (*Figure 6D*), thereby reducing the overall rate of FC production relative to *UAS-Hop* FSC lineages. Accordingly, labeled FSCs accumulated to an even greater extent than for *UAS-Hop* alone and indeed exceeded the normal capacity of the germarium at 28.1 *axn UAS-Hop* FSCs per germarium (*Figure 6H–J*). Thus, changes in Wnt pathway activity profoundly altered the competitive success of FSCs with elevated JAK-STAT pathway activity by either promoting (loss of Wnt) or reducing (elevated Wnt) FC production, without significantly altering FSC division rates in either case.

## Discussion

Our investigations have generated a detailed picture of how a variety of fundamental stem cell behaviors, including precise location, division, differentiation, survival and amplification, respond to positional cues relayed by the activity levels of two major signaling pathways that are graded with complementary polarities across the stem cell domain. The results reveal JAK-STAT pathway activity as the primary dose-dependent agent dictating the pattern of FSC proliferation and as a major influence promoting conversion of FSCs to FCs. Wnt pathway magnitude relative to neighboring FSCs was previously shown to be a major determinant of the AP position of an FSC and its conversion to an EC (*Reilein et al., 2017*). Here we found that elevated Wnt pathway activity also inhibits FC production while elimination of Wnt pathway activity promotes conversion of posterior FSCs to FCs. These and other results suggest an outline for how external signals specify a functional stem cell domain and some elements of the mechanisms for coordinating division and differentiation in a fluid collection of instantaneously heterogeneous stem cells maintained by population asymmetry.

The original perception of just two rigidly held FSCs per germarium, repeatedly undergoing asymmetric divisions to produce FCs (*Margolis and Spradling, 1995*) was replaced by a very different picture of 14–16 mobile FSCs of varied lifetimes, division rates and two alternative differentiation products (FCs and ECs) on the basis of detailed examination of FSC lineages over a variety of time periods (*Hayashi et al., 2020*; *Reilein et al., 2017*). The original model provided no basis for explaining why FSC maintenance was dependent on FSC division rate, why both increased and

decreased Wnt signaling led to stem cell loss, or why FSC maintenance relied on input from almost every major signaling pathway tested. The revised model, in which FSC division and differentiation are independent, explains why the division rate of one stem cell relative to others is a key determinant of FSC competition (*Reilein et al., 2018*). Likewise, we now understand that FSC loss is driven by excessive conversion of FSCs to ECs in response to elevated Wnt pathway activity (*Reilein et al., 2017*) and by excessive conversion of FSCs to FCs in response to loss of Wnt pathway activity (this study). Now that we understand that FSC behavior has many independent facets that are potentially subject to regulation, including precise AP location, rate of division and differentiation to ECs or FCs, we are also beginning to understand the initially surprising finding that normal FSC behavior depends on the activities of multiple signaling pathways (*Castanieto et al., 2014*; *Johnston et al., 2016*; *Kirilly et al., 2005*; *O'Reilly et al., 2008*; *Reilein et al., 2017*; *Song and Xie, 2003*; *Vied et al., 2012*; *Wang et al., 2012*; *Zhang and Kalderon, 2001*). Importantly, in this study, we found that the measured overall maintenance and amplification of an FSC is consistent with the sum of the individual behavioral responses we measured in response to a large variety of changes in two of the major signaling pathway activities, in isolation and in various combinations, affirming the validity of our overall conception of FSC behavior.

Our findings provide insights of unprecedented detail of how a stem cell community maintained by population asymmetry can be regulated by spatially-restricted signals and can therefore inform future studies of stem cells with a similar organization, including mammalian intestinal stem cells. Some of the genetic conditions we examined also provide clear precedents for how different sets of mutations might cause diseases related to either cancers initiated in stem cells or stem cell dysfunction.

## FSC proliferation

The near-parallel gradients of FSC division frequency reported by EdU incorporation and JAK-STAT pathway activity, declining from posterior to anterior across the FSC domain suggested a potential causal link (*Figure 7*). Indeed, when the JAK-STAT pathway was globally manipulated to be uniform across the EC and FSC domains, at a level marginally higher than seen normally in posterior FSCs, all FSCs were seen to incorporate EdU at the same high frequency and even some cells in EC locations entered the cell cycle. Thus, the pattern of JAK-STAT pathway activity appears to be a key determinant of the pattern of FSC divisions (*Figure 7*). Moreover, the anterior border of dividing somatic cells, a key characteristic of active stem cells, appears to be set by JAK-STAT pathway activity dropping below a critical threshold.

In our studies EdU was incubated with freshly dissected ovarioles for 1 hr, so the EdU index reports the fraction of cells in S phase at the time of dissection or shortly afterwards. A higher EdU index reports a higher proportion of cells in S phase, which commonly correlates with a higher rate of cycling of the cell population assayed, but it is only a quantitative measure of the rate of cell division if the length of S phase is unchanged. It is quite possible that the length of S phase may differ between FSCs in different AP locations or under different genetic conditions, so the EdU labeling indices we report provide an indication, rather than a definitive measure of FSC division rates. Thus, the uniform EdU incorporation observed when JAK-STAT signaling was made uniform may not report the true status of FSC division rates.

Additional influences on the FSC division pattern were clearly revealed in the absence of JAK-STAT pathway activity by a robust AP gradient of EdU incorporation observed in five different genotype combinations covering a range of overall FSC division rates. We cannot yet tell whether the signals responsible for that polarized behavior normally augment JAK-STAT pathway action or serve only as a latent reserve system. Graded Hh signaling strongly promotes FSC division but it declines from anterior to posterior and cannot therefore underlie the converse gradient of FSC division (*Hartman et al., 2013*; *Hartman et al., 2010*; *Huang and Kalderon, 2014*; *Vied et al., 2012*). Wnt pathway activity clearly intersects with the mechanisms that regulate FSC cell cycles but it does not appear to have a major influence on the AP pattern of FSC divisions. Under conditions of normal JAK-STAT pathway activity, increased Wnt signaling strongly inhibits FSC division but global reduction of Wnt pathway activity that drastically diminishes graded signaling, permitted a normal pattern of FSC EdU incorporation at roughly normal frequencies. Similarly, reduced Wnt pathway activity strongly increased FSC division when STAT was inactivated but those FSCs with diminished Wnt signaling still showed a normal AP gradient of EdU incorporation. A potential physiological focus of

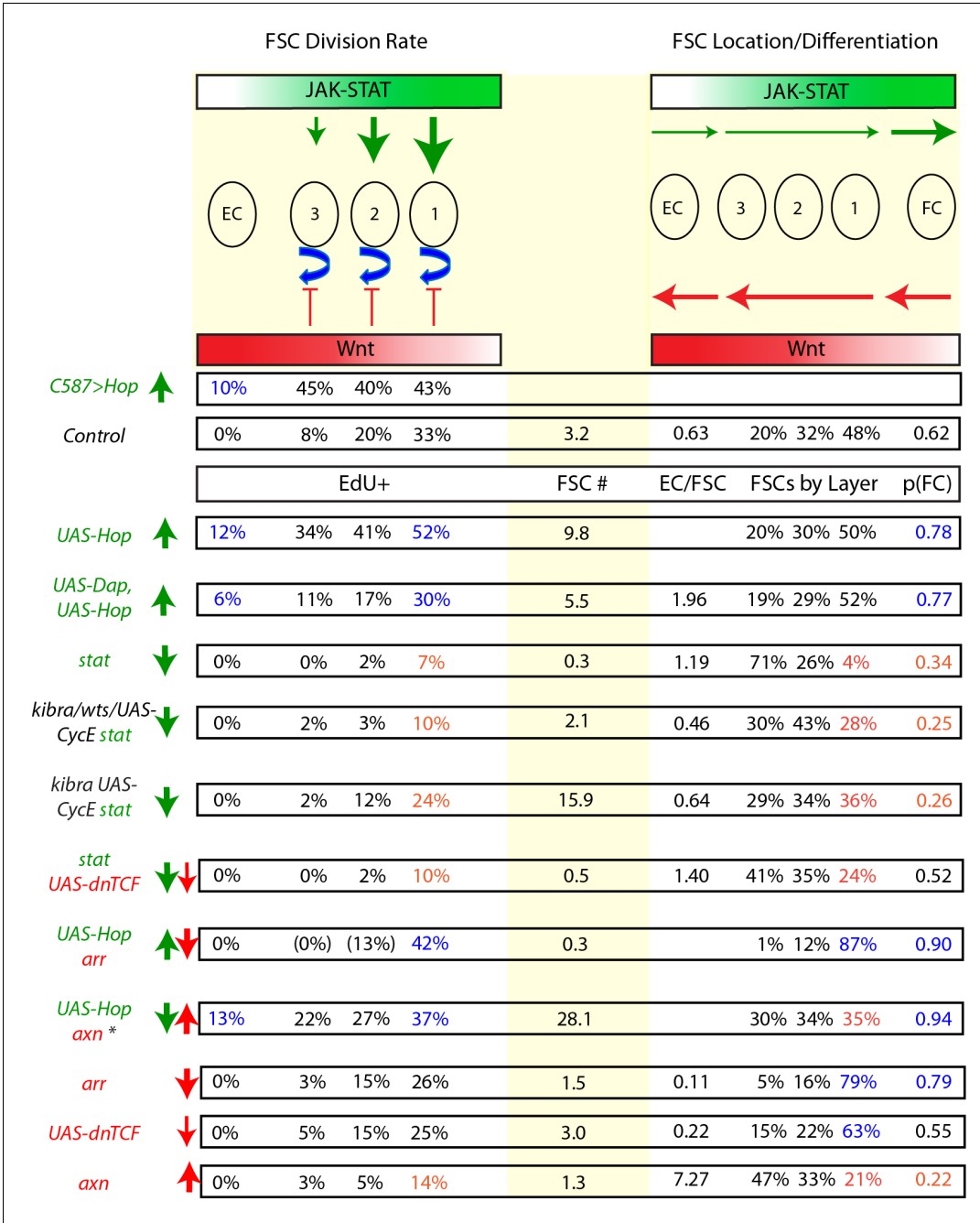

**Figure 7.** Summary of the cell autonomous influences of JAK-STAT and Wnt pathway magnitudes on FSC behavior. Diagrammatic summary of key results supporting inferences of how graded JAK-STAT (green shading) and Wnt (red shading) pathways regulate the division rates of ECs and FSCs in layers 1–3 (left) and the location of FSCs among layers 1–3, the conversion of anterior FSCs to ECs and posterior FSCs to FCs (right). The central column shows the average number of marked FSCs per germarium at 12d in MARCM tests. Genotypes at the extreme left are paired with vertical arrows indicating the direction (up/down) and magnitude (thickness) of JAK-STAT (green) and Wnt (red) pathway alterations. For the *C587>Hop* experiment only division rates were measured as the fraction of all FSCs in a given layer, or all ECs, incorporating EdU. Other tabulated values report the properties of marked cells in MARCM clones. Asterisk for *axn UAS-Hop* indicates results are for the test at 22C. Measurements of division rate in the EC region or any layer 1 FSC behavior (division rate, proportion of all FSCs or conversion to FCs) are highlighted by colored text if there was a notable increase (blue) or decrease (orange). JAK-STAT signaling promotes FSC division in proportion to graded pathway activity (green vertical arrows, top), while Wnt pathway activity only inhibits division (red verticals) under artificial conditions of increased Wnt signaling

*Figure 7 continued on next page*

*Figure 7 continued*

or elimination of JAK-STAT pathway activity. FC production from posterior FSCs is promoted by JAK-STAT (horizontal green arrow) and by elimination of Wnt pathway activity (reverse horizontal red arrow). Consequently, *stat* mutant FSCs were maintained better than expected from their greatly reduced division rates but FSCs with high JAK-STAT activity and no Wnt pathway activity were rapidly drained despite high division rates. Increased Wnt pathway activity always favored more anterior FSC locations and EC production (12d EC/aFSC values are shown) from anterior FSCs (horizontal red arrows), and generally had a stronger influence than the opposing influence of JAK-STAT in anterior regions (thinner green reverse horizontal arrows). By contrast, increased JAK-STAT activity overcame the influence of increased Wnt pathway activity on conversion of FSCs to FCs at the posterior of the FSC domain. Values for EdU in parentheses for *arr UAS-Hop* indicate unreliable values because only a very low number of FSCs were present in those locations (n < 10).

Wnt pathway influence is at the EC/FSC border where Wnt pathway activity is high and JAK-STAT pathway low. While we did not observe any ectopic cell division in the EC region in response to Wnt pathway reduction or elimination, those observations were limited by the low frequency of Wnt pathway deficient cells in EC locations and the possibility that such cells may be prone to apoptosis or other stresses. Hence, it remains possible that high Wnt pathway activity may contribute to defining the anterior border of dividing somatic cells.

## FSC location and differentiation

Through concerted actions on two separable parameters of behavior (FSC AP location and conversion of posterior FSCs to FCs), FC production is promoted by JAK-STAT pathway activity and reduced by Wnt pathway activity, in keeping with the relative levels and gradients of these pathways in the posterior half of the FSC domain (*Figure 7*). Although the mechanisms for FSC to FC conversion are currently unknown, it involves association with germline cysts and the beginning of a transition to an epithelial phenotype, neither of which are apparent in the AP movements of FSCs between layers. The concerted actions of Wnt and JAK-STAT pathways on both FC production and FSC AP location are not therefore likely different manifestations of exactly the same molecular responses to signaling. The responses to loss of Wnt signaling and elevated JAK-STAT pathway activity on these behaviors were found to be additive, with that specific combination causing an extremely high rate of FC production, severely depleting the marked FSC pool. In effect, that result shows that the signaling environment posterior to layer 1 FSCs cannot support maintenance of an FSC.

EC production from anterior FSCs was strongly favored by increasing Wnt pathway activity, as noted previously (*Reilein et al., 2017*), and was increased by loss of JAK-STAT signaling. We also found evidence of significant turnover of wild-type ECs produced from FSCs during adulthood and the limited effects of blocking apoptosis suggest that a major component of turnover may be a return of those cells to the FSC domain. Our measurements did not separate EC production from EC turnover but it seems likely that the changes we observed in EC accumulation in response to altered Wnt and JAK-STAT signaling may be due to regulation of both processes.

## Co-ordination of FSC responses to external signals

By examining twin-spot products of recombination in an FSC (with complementary colors) where one daughter lineage consisted of just a single FC patch, it was possible to measure the time between division of an FSC and acquisition of FC status, revealing that an FSC can become an FC at any time after its last division (*Reilein et al., 2018*). Thus, for an individual FSC, division and differentiation to an FC are separate processes. That separation of fundamental stem cell activities allows the possibility of independent regulation of each process at the single-cell level. There may, nonetheless, be some systematic connections among individual, separable FSC behaviors that coordinate behavior at the single-cell level.

Our results to date suggest that most FSC behaviors are largely independent. One exception was a potentially systematic connection between division rate and AP location. We found that *cycE* mutant FSCs, which may have reduced division as the only direct consequence of the mutation, had a small anterior bias and that the anterior bias of *stat* mutant FSCs was significantly reduced when division rate was increased by elevating CycE expression or Yki activity. Changes in AP location

might plausibly be caused by poorer competition of slow-dividing FSCs in the posterior layer where normal FSCs divide and become FCs at a high rate.

In contrast to the noted effect of division rate on AP location, the greatly reduced FSC to FC conversion of *stat* mutants was not altered by increasing division rate through additional genetic manipulations, and the increased conversion of FSCs to FCs due to elevated JAK-STAT pathway activity was not altered by reducing FSC division rates (with a Cdk2 inhibitor). Thus, the mechanisms that regulate FSC division and FSC differentiation do not appear to be robustly coupled within single cells. However, the balance between these two processes might in theory still be achieved in single stem cells if a key regulatory signal affected both division and differentiation in appropriate proportions.

The JAK-STAT pathway was found to promote both FSC division and conversion to FCs, while reducing net conversion to ECs. About 5–6 FSCs become FCs per 12 hr budding cycle, while only one fourth as many FSCs become ECs (*Reilein et al., 2018*). Conversion to FCs is therefore the main drain on the FSC population that mandates compensatory FSC division. Because JAK-STAT signaling stimulates both FSC division and conversion to FCs in a dose-dependent manner, there will be a tendency to maintain each FSC, without loss or amplification, even when there are stochastic or systemic changes in the strength of this pathway. Also, by instructing posterior FSCs to divide faster than anterior FSCs the generation and loss of FSCs is roughly balanced for each layer, allowing for a roughly equal dynamic exchange of FSCs between layers, rather than, for example, a net anterior to posterior flow that would make anterior FSCs systematically longer-lived. The role of JAK-STAT signaling coordinating FSC division and differentiation was evident when pathway activity was eliminated and only the division rate of *stat* mutant FSCs was elevated by genetic alteration of other agents (in *kibra UAS-CycE stat* FSCs). The consequence was a large accumulation of hyper-competitive marked FSCs that supported very little FC production (*Figure 7*). Graded Wnt pathway activity appears to be important for supplementing the cell autonomous coordinating activity of JAK-STAT signaling. When JAK-STAT pathway is elevated, increased FSC division is partially offset by increased FC production and marked FSCs amplify over time. FSC amplification was greatly accelerated when Wnt pathway activity was genetically increased (in *axn UAS-Hop* FSCs) because FSC division remained rapid but FC production was limited through FSCs adopting more anterior locations (*Figure 7*). By contrast, when Wnt pathway activity was eliminated (in *arr UAS-Hop* FSCs) the marked FSC population was drained rapidly because conversion to FCs was enhanced without markedly altering FSC division rates (*Figure 7*).

Although dual JAK-STAT pathway responses, supported by Wnt pathway input to FSC AP location and conversion to FCs, contribute cell autonomously to balancing FSC division and differentiation, as described above, that balance is largely exercised at the community level when stem cells are maintained by population asymmetry. Several parameters must be balanced at the community level. The average FSC division rate must be appropriate to support production of a specific number of FCs and ECs every 12 hr but this also depends on the total number of FSCs. This, in turn, appears to be defined by a specific domain or space where FSCs can reside.

A key general question is how a stem cell domain is spatially-defined. The FSC paradigm provides an example where this can be understood in terms of the distribution and influences of ligands for two major extracellular signaling pathways. Both graded JAK-STAT and Wnt pathways contribute to both borders. The anterior border is between non-dividing ECs and FSCs. Increasing Wnt pathway activity and decreased JAK-STAT pathway activity both promote FSC to EC conversion, while declining JAK-STAT signaling also limits the proliferative zone. The posterior border is between FSCs and FCs. Increased JAK-STAT and reduction of Wnt pathway activity (to zero) promote the FSC to FC transition. Thus, the graded nature of both pathways plays a crucial role in determining the A/P extent of the FSC domain and, consequently, the number of FSCs supported.

The influences of JAK-STAT and Wnt pathways were examined here as cell autonomous responses in mosaic tissues (where most cells have normal genotypes). Further experiments manipulating pathway activities globally (in all cells) may well result in a number of compensatory changes in behavior and signaling properties, and will have to be evaluated in detail to understand to what extent the size and location of the FSC domain depends on the normal magnitude and gradations of these and other pathways.

The organization of mammalian intestinal stem cells is quite similar to FSCs and Wnt signaling also plays a prominent role. There, Wnt signals derive from mesenchymal cells surrounding the crypt

base and additionally from Paneth cells in the small intestine, to form a gradient (*Gehart and Clevers, 2019*). In collaboration with R-Spondin signals, the magnitude of Wnt signaling appears to be translated into a variety of responses, including promotion of stem cell division and the expression of a key transcription factor, Ascl2, and Ephrin signaling components, which contribute to regulating cell migration and the transition to transit-amplifying cells (*Batlle et al., 2002*; *Gehart and Clevers, 2019*; *Schuijers et al., 2015*; *Yan et al., 2017*). Whether stem cell division is patterned and whether additional spatially-restricted signals guide the development of different stem cell products remain to be explored. In the FSC paradigm the dependence of stem cells on intermediate, rather than the highest levels of Wnt signaling as in the mammalian intestine, and the use of a second major graded patterning influence may be essential adaptations to differentiation occurring at two faces of the stem cell domain.

## Verification of the organization of FSCs

FSCs (originally termed SSCs) were first identified by lineage studies, which noted that long-lasting lineages generally extended from roughly the mid-point of the germarium (*Margolis and Spradling, 1995*). The conclusions from the same study that there were just two FSCs per germarium, each with a half-life around two weeks were subsequently cited many times. However, this dogma was dramatically revised two decades later through an extensive series of lineage studies comprising three different approaches to assess the number of FSCs as 14–16, the first definitive approach to identifying precise FSC locations and a demonstration that FSCs produced ECs as well as FSCs. A key aspect of this revision was the realization that earlier studies had implicitly assumed that FSCs are long-lived and maintained by single-cell asymmetry, leading to the exclusion of the majority of FSC lineages from all analyses, necessarily leading to substantial under-estimation of FSC numbers, over-estimation of average FSC longevity and obscuring the evidence that FSCs are in fact maintained by population asymmetry. Our current picture of FSCs is supported by extensive direct evidence; recent claims supporting the original conception of FSCs (*Fadiga and Nystul, 2019*) were addressed, and in our opinion, refuted comprehensively (*Kalderon, 2020*). Nevertheless, additional evidence can always contradict, confirm or refine an existing model.

Here we have presented extensive quantitative sets of data concerning FSC division, location and differentiation. A demanding test of the FSC model is whether the aggregation of these measured parameters fits well with the measured retention, loss or expansion of FSCs of a large range of mutant genotypes, which instruct a wide range of altered behaviors. We found that this was the case, as discussed above. One or two outcomes are especially noteworthy with regard to the number and location of FSCs. When FSCs lacked STAT activity but were stimulated by additional genetic changes to divide at rates approaching normal FSCs, those *kibra UAS-CycE stat* FSCs were present at an average of 15.9 FSCs per germarium, including multiple cells in each FSC layer (*Figure 3A,E, H*). The layer 1 FSCs were clearly stem cells rather than FCs because this genotype rarely produced FCs captured in any location along the ovariole (*Figure 3E*). Cells in layers 2 and 3 were also clearly stem cells rather than ECs because a large fraction incorporated EdU at a given time, while ECs are naturally quiescent. The phenotype of *arr* mutant FSC lineages, where almost all labeled cells are in layer one or further posterior (*Figure 4A*) provides further evidence that layer 1 cells are stem cells because these *arr* mutant lineages survive almost as well as control lineages (*Figure 6J*). These observations provide further confirmation of our current FSC model and are clearly not consistent with either the original models postulating just two FSCs in a single layer (*Fadiga and Nystul, 2019*; *Margolis and Spradling, 1995*; *Nystul and Spradling, 2007*; *Nystul and Spradling, 2010*) or recent postulates of an intermediate model where FSCs occupy the full circumference of the germarium but only in layer 2 (*Singh et al., 2018*).

Some of the mutant FSC phenotypes we observed also illustrate disease-relevant situations that could arise in other stem cells with a similar organization, such as mammalian intestinal stem cells. The relevance of mutations that allow amplification of specific stem cell genotypes to cancer origins has been described previously, both within the concept of field cancerization and specifically with regard to the effect of mutations conferring increased rates of stem cell division when differentiation is not coupled to stem cell division (*Frede et al., 2014*; *Reilein et al., 2018*; *Ritsma et al., 2014*; *Rompolas et al., 2016*; *Slaughter et al., 1953*; *Vermeulen and Snippert, 2014*). In those situations, exemplified by *ptc* mutations activating the Hh pathway in FSCs, a mutant stem cell expands within the normal stem cell domain to provide a stable, expanded source of hyperproliferative stem cell

derivatives (*Huang and Kalderon, 2014*; *Reilein et al., 2018*; *Vied and Kalderon, 2009*). The properties of FSCs with increased JAK-STAT signaling provide an illustration of a more potent variant of the same principle because the genetic alteration also allows expansion of the proliferative domain into more anterior EC territory. Interestingly, for FSCs there is a genetic remedy; loss of Wnt signaling encouraged differentiation of these hyperproliferative stem cells and effectively extinguished the mutant lineages. Finally, the *kibra UAS-CycE stat* genotype provides a strikingly different paradigm. Here, the mutant stem cells also amplify but these stem cells do so because they rarely differentiate; they are not hyper-proliferative. If these differentiation-defective stem cells eventually out-compete all normal stem cells the ability of the stem cell community to support continued production of derivative cells will be severely compromised.

# Materials and methods

Key resources table

| Reagent type (species) or resource | Designation | Source or reference | Identifiers | Additional information |
|---|---|---|---|---|
| Gene (*D. melanogaster*) | cycE | Flybase ID: FBgn0010382 | CG3938 | |
| Gene (*D. melanogaster*) | cutlet | Flybase ID: FBgn0015376 | CG33122 | |
| Gene (*D. melanogaster*) | hop | Flybase ID: FBgn0004864 | CG1594 | |
| Gene (*D. melanogaster*) | dap | Flybase ID: FBgn0010316 | CG1772 | |
| Gene (*D. melanogaster*) | pan (TCF) | Flybase ID: FBgn0085432 | CG34403 | |
| Gene (*D. melanogaster*) | sha | Flybase ID: FBgn0003382 | CG13209 | |
| Gene (*D. melanogaster*) | arr | Flybase ID: FBgn0000119 | CG5912 | |
| Gene (*D. melanogaster*) | stat | Flybase ID: FBgn0016917 | CG4257 | |
| Gene (*D. melanogaster*) | axn | Flybase ID: FBgn0026957 | CG7926 | |
| Gene (*D. melanogaster*) | apc1 | Flybase ID: FBgn0015589 | CG1451 | |
| Gene (*D. melanogaster*) | apc2 | Flybase ID: FBgn0026598 | CG6193 | |
| Gene (*D. melanogaster*) | kibra | Flybase ID: FBgn0262127 | CG33967 | |
| Gene (*D. melanogaster*) | wts | Flybase ID: FBgn0011739 | CG12072 | |
| Genetic Reagent (*D. melanogaster*) | hs-flp | PMID:7867064 | FBti0002738 | hsp70-driven Flp recombinase on X |
| Genetic Reagent (*D. melanogaster*) | NM FRT40A | BDSC BL-1835 | | Control for MARCM clones |
| Genetic Reagent (*D. melanogaster*) | FRT42D ubi-GFP | BDSC BL-5626 | | Control for MARCM clones |
| Genetic Reagent (*D. melanogaster*) | FRT82B NM | PMID:23079600 | | Control for MARCM clones |
| Genetic Reagent (*D. melanogaster*) | C587-GAL4 | BDSC BL-67747 | FBti0037960 | GAL4 expressed in ECs and FSCs |
| Genetic Reagent (*D. melanogaster*) | UAS-dnTCF | BDSC BL-4784 BDSC BL-4785 | FBtp0012500 | Dominant-negative TCF on 2nd (4784) and 3rd (4785) chromosome |

*Continued on next page*

Continued

| Reagent type (species) or resource | Designation | Source or reference | Identifiers | Additional information |
|---|---|---|---|---|
| Genetic Reagent (D. melanogaster) | UAS-dap | PMID:10790398 | FBtp0001369 | Inhibitor of CycE/Cdk2 |
| Genetic Reagent (D. melanogaster) | UAS-Hop$^{3W}$ | PMID:23079600 | | Activated form of Hopscotch (JAK) |
| Genetic Reagent (D. melanogaster) | UAS-DIAP1 | PMID:22473013 | | Inhibitor of apoptosis (3$^{rd}$ chromosome) |
| Genetic Reagent (D. melanogaster) | UAS-CycE | PMID:19966222 | | GAL4-responsive CycE |
| Genetic Reagent (D. melanogaster) | FRT42D tub-GAL80 | PMID:28414313 | | For 2R MARCM clones |
| Genetic Reagent (D. melanogaster) | FRT82B tub-GAL80 | BDSC BL-5135 | | For 3R MARCM clones |
| Genetic Reagent (D. melanogaster) | Fz3-RFP | PMID:28414313 | | Wnt pathway activity reporter; used on 2$^{nd}$ and 3$^{rd}$ chromosome |
| Genetic Reagent (D. melanogaster) | STAT-GFP | PMID:23079600 | | JAK-STAT pathway reporter |
| Genetic Reagent (D. melanogaster) | STAT-RFP | PMID:31140975 | | JAK-STAT pathway reporter |
| Genetic Reagent (D. melanogaster) | cycE$^{WX}$ | PMID:19966222 | FBal0241968 | hypomorphic allele |
| Genetic Reagent (D. melanogaster) | cutlet$^{4.5.43}$ | PMID:22473013 | | |
| Genetic Reagent (D. melanogaster) | arr$^2$ | PMID:23079600 | FBal0000724 | amorphic allele |
| Genetic Reagent (D. melanogaster) | stat$^{85C9}$ | PMID:23079600 | FBal0130583 | amorphic allele |
| Genetic Reagent (D. melanogaster) | stat$^{06346}$ | PMID:23079600 | FBal0009559 | amorphic allele |
| Genetic Reagent (D. melanogaster) | axn$^{E77}$ | PMID:28414313 | FBal0121005 | Q406 stop codon, likely null |
| Genetic Reagent (D. melanogaster) | axn$^{SO44320}$ | PMID:28414313 | FBal0097414 | enhancer trap null |
| Genetic Reagent (D. melanogaster) | apc1$^{Q8}$ | PMID:28414313 | FBal0091898 | Q427 stop codon |
| Genetic Reagent (D. melanogaster) | apc2$^{D40}$ | PMID:28414313 | FBal0137655 | hypomorphic allele |
| Genetic Reagent (D. melanogaster) | kibra$^{32}$ | PMID:24798736 | FBal0244965 | deletion in 5' UTR and first exon |
| Genetic Reagent (D. melanogaster) | kibra$^{del}$ | PMID:24798736 | FBal0244407 | amorphic allele |
| Genetic Reagent (D. melanogaster) | wts$^{x1}$ | PMID:24798736 | FBal0044527 | amorphic allele |
| Chemical compound, drug | Normal goat serum | Jackson ImmunoResearch Laboratories | 005-000-121 | 10% in PBS for blocking |
| Antibody | anti-GFP (rabbit polyclonal) | Molecular Probes | A6455 | (1:1000) |
| Antibody | anti-Fas3 (mouse monoclonal) | DSHB | 7G10 | (1:250) |
| Antibody | AlexaFluor 488, 546, 594, 647 | Thermofisher Scientific | | (1:1000) |

*Continued*

| Reagent type (species) or resource | Designation | Source or reference | Identifiers | Additional information |
| --- | --- | --- | --- | --- |
| Chemical compound, drug | DAPI Fluoromount-G | Southern Biotech | 0100–20 | Mount samples and stain for DAPI |
| Commercial assay or kit | Click-iT EdU Cell Proliferation Kit | ThermoFisher Scientific | C10086 | |
| Software, algorithm | ZEN Blue, ZEN Black, and ZEN Lite | Zeiss | | For viewing Z-stack images and quantifying fluorescence |

## MARCM clonal analysis

1-3d old adult *D. melanogaster* females with the appropriate genotypes were given a single 30 min (for *FRT40A* and *FRT42D*) or 45 min (for *FRT82B*) heat shock at 37C, with the heat shock duration determined by the observed relative rate of recombination for different *FRT* sites. Afterwards, flies were incubated at 25°C, 29°C or 22°C. Higher temperature increases GAL4 activity and was used for expression of *UAS-Hop* when using *FRT40A* or *FRT42D*, while 22°C was also used for the *axn UAS-Hop* genotype to moderate activity. Flies were maintained by frequent passage on normal rich food supplemented by fresh wet yeast during the 12d experimental period. Flies were dissected at 6d and 12d. We waited until 6d to ensure that all GAL80 present in cells, prior to clone induction, would be titrated out, permitting robust GAL4 induction of *UAS-GFP* and any additional transgenes. The 6d time point also ensured that any marked cells are derived from FSCs, as dividing FCs marked in the heat shock would have passed through the ovariole in less than 5d.

Immediately after dissection, 6d ovaries underwent 1 hr of EdU labelling based on the protocol of the Click-iT Plus EdU Cell Proliferation Kit for Imaging (Invitrogen). Both 6d and 12d ovaries were stained for Fasciclin III (Fas3) and GFP. Ovaries were then manually separated into constituent ovarioles, and mounted using DAPI Fluoromount-G (SouthernBiotech) to stain nuclei. Ovarioles were imaged with a Zeiss LSM700 or LSM800 confocal microscope, operated in part by the Zeiss ZEN software. The entire germarium was captured in the images, as well as an average of 3–4 egg chambers. Collected images were saved as CZI files, and were later analyzed utilizing the ZEN Lite software. We aimed to image 50 germaria for every genotype in each experiment.

## MARCM genotypes

Flies with alleles on an *FRT40A*, *FRT42D*, or *FRT82B* chromosomes were used in MARCM experiments using the following genotypes:

*FRT40A*: *yw hs-Flp, UAS-nGFP, tub-GAL4/yw; act-GAL80 FRT40A / (X)FRT40A; act >CD2>GAL4/UAS-(Y)* – where X, Y combinations included: (X) – *NM* (Nuclear Myc, Control), *cycE^{WX}*, *cutlet^{4.5.43}* (Y) - *UAS-Hop^{3W}, UAS-Dap, UAS-dnTCF*.

*FRT42D*: *yw hs-Flp, UAS-nGFP, tub-GAL4/yw; FRT42D act-GAL80 tub-GAL80/FRT42D (X); act >CD2>GAL4/UAS-(Y)* – where X, Y combinations included: (X) – *sha* (Control), *ubi-GFP* (Control), *arr^2*, (Y) – *UAS-Hop^{3W}, UAS-DIAP1, Fz3-RFP*.

*FRT82B*: *yw hs-Flp, UAS-nGFP, tub-GAL4/yw; act >CD2>GAL4 UAS-GFP/UAS-(Y); FRT82B tub-GAL80/FRT82B (X)* – where X,Y combinations included: (X) – *NM* (control), *stat^{85C9}, stat^{06346}, axn^{E77}, axn^{S044320}, apc1^{Q8}apc2^{D40}, kibra^{32}, kibra^{del}, wts^{x1}, UAS-Hop^{3W}, UAS-CycE, UAS-dnTCF*, (Y) *UAS-dnTCF, Fz3-RFP, STAT-RFP*, including combinations of (X) elements.

All tests not involving *UAS-Hop* were performed at 25°C. For *UAS-Hop* in experiments with *FRT40A* or *FRT42D* a temperature of 29°C was required to observe strong excess JAK-STAT phenotypes, as reported previously (*Vied et al., 2012*) (at 29°C *act-GAL80* [provided by T. Laverty, Janelia Farms] in place of *tub-GAL80* on 2L, or in addition to *tub-GAL80* on 2R in MARCM clone stocks were essential to suppress GFP expression in non-recombined cells). The *UAS-Hop* insertion is on 3R, so that clones made with *FRT82B* contain two copies of *UAS-Hop*, with tests performed at 25°C. The phenotypes due to two copies at 25°C were overlapping with those from one copy at 29°C and those data were aggregated in summary results presented. For *FRT82B* clones with *axn* and *UAS-Hop* on 3R the accumulation of marked cells was so high at 25°C that samples could not be scored

reliably at 12d after heat shock. Consequently, additional tests for that genotype, *FRT82B Hop* and an *FRT82B* control were performed in parallel at 22°C, with results given separately from those obtained at 25°C.

### C587-GAL4 experiments

1-3d old flies of the genotype *C587-GAL4; UAS-X/ts-GAL80, FRT42D tub-lacZ; (Reporter)/TM6B* were chosen, where *UAS-X* was *UAS-dnTCF*, *UAS-CycE*, or *UAS-Hop*, and the reporter was either STAT-GFP or Fz3-RFP. Flies were incubated at 29°C for 3d, and *UAS-Hop* flies were also incubated for 6d and 10d. Dissected ovaries underwent the EdU and Immunohistochemistry protocols as above, without staining for GFP. For Fz3-RFP experiments with EdU, Alexa Fluor 488 dye was used instead of 594 to avoid spectral overlap.

### EdU protocol

Ovaries were directly dissected into a solution of 15 μM EdU in Schneider's *Drosophila* media (500 μl, Gibco) and incubated for one hour at room temperature. These tubes were laid on their side and rocked manually, to ensure all dissected ovaries were fully submerged. Ovaries were then fixed in 3.7% paraformaldehyde in PBS for 10 min, treated with Triton in PBS (500 μl, 0.5% v/v) for 20 min, and rinsed 2x with bovine serum albumin (BSA) in PBS (500 μl, 3% w/v) for 5 min each rinse. Ovaries were exposed to the Click-iT Plus reaction cocktail (500 μl) for EdU visualization, for 45 min. The reaction cocktail was freshly prepared prior to use, with reagents from the Invitrogen Click-iT Plus EdU Cell Proliferation Kit for Imaging, including the Alexa Fluor 594 dye. Ovaries were then rinsed 3x with BSA in PBS (500 μl, 3% w/v) for 5 min each rinse.

### Immunohistochemistry

For experiments without EdU, ovaries were dissected directly into a fixation solution of 4% paraformaldehyde in PBS for 10 min at room temperature, rinsed 3x in PBS, and blocked in 10% normal goat serum (NGS) (Jackson ImmunoResearch Laboratories) in PBS with 0.1% Triton and 0.05% Tween-20 (PBST) for 1 hr. Monoclonal antibodies for Fas3 were obtained from the Developmental Studies Hybridoma Bank, created by the NICHD of the NIH and maintained at The University of Iowa, Department of Biology, Iowa City, IA 52242. 7G10 anti-Fasciclin III was deposited to the DSHB by Goodman, C. and was used at 1:250 in PBST. Other primary antibodies used were anti-GFP (A6455, Molecular Probes) at 1:1000 in PBST. Ovaries were incubated in primary antibodies overnight, rinsed three times in PBST, and incubated 1–2 hr in secondary antibodies Alexa-488 and Alexa-647 (ThermoFisher) at 1:1000 in PBST to label GFP and Fas3, respectively. DAPI-Fluoromount-G (Southern Biotech) was used to mount ovaries.

### Imaging and scoring

All germaria were imaged in three dimensions on an LSM700 or LSM800 confocal laser scanning microscope (Zeiss) and using a 63 × 1.4 N.A. lens. Zeiss ZEN software was used to operate the microscope and view images. Images were typically 700 × 700 pixels with a bit depth of 12. The scaling per pixel was 0.21 x 0.21 x 2.5 μm. The range indicator in ZEN was used to determine the appropriate laser intensity and gain. ZEN was used to linearly adjust channel intensity for dim signals to improve brightness without photobleaching samples. Images were saved as CZI files and scored directly in ZEN. DAPI and Fas3 staining were used as landmarks to guide scoring. Marked cells were considered FSCs if they were within three cell diameters anterior of the Fas3 border. Cells immediately adjacent to the border were considered to be in Layer 1, with Layers 2 and 3 in sequentially anterior positions. Anterior to the FSC niche, the EC region was roughly divided into two halves, with region 2a ECs immediately anterior to FSCs and region 1 ECs anterior to that. Germaria were also scored (Y/N) for the presence of marked FCs. For the 'immediate FC' method tabulation, the presence of an FC immediately posterior to Layer one was also scored Y/N. For publication, images were digitally zoomed in ZEN and exported as tif files using the 'Contents of Image Window' function. Images were rotated in Abode Photoshop CS5 to uniformly orient the germaria.

### Measurement of signaling pathway reporter activities

STAT-GFP, STAT-RFP and Fz3-RFP reporter activity was quantified within ZEN software. Using the Draw Spline Contour function, an outline of a DAPI cell nuclei was traced, and the fluorescence intensity within the outline was recorded. The outline of cells not expressing the reporter strongly (anterior ECs for JAK-STAT reporters and FCs for Fz3-RFP) were used to determine background intensity and were subtracted from calculated totals. For quantification of signaling pathway gradients in experiments using *C587-GAL4* the signal in germline cells of the first egg chamber was used to determine background intensity. Also, the intensity of FCs from the second or third egg chamber of each sample was used as a reference, and all intensity measurements of cells within the germarium were divided by the reference to produce a relative intensity that could be compared between samples. For quantification of individual clones, the RFP intensity of a GFP-positive cell was divided by that of a GFP-negative cell in a similar position along the A/P axis, within the same or an adjacent Z plane and an average was calculated from many such pairs to derive the percentage intensity for labeled cells relative to unlabeled cells.

### Statistics and reproducibility

All images shown are representative of at least ten examples. In most cases the number is much higher and is given explicitly where relevant for statistical analysis of outcomes. No statistical method was used to predetermine sample size but we used prior experience to establish minimal sample sizes. No samples were excluded from analysis, provided staining was of high quality. The experiments were not randomized; all samples presented as groups in the results were part of the same experiment and treated in exactly analogous ways without regard to the identity of the sample. Investigators were not blinded during outcome assessment, but had no pre-conception of what the outcomes might be. For EdU incorporation, FSC layers, Immediate FC probability tabulations, proportion of germaria with FSCs and/or FCs, the 'N-1' Chi-squared test method was used to calculate a Z score for determining significance between indicated genotypes, and error was reported as standard error of a proportion. To determine whether the EdU index distribution among the FSC layers of an altered genotype different from controls, we first calculated the average EdU index for all FSCs of the altered genotype, with each layer contribution weighted based on the normal distribution of FSC among layers measured in appropriate controls (for MARCM or *C587-GAL4* tests). This average EdU index was then multiplied by the control EdU index for each layer to derive expected EdU indexes for each layer of an altered genotype if the EdU pattern matched controls. Finally, a chi-squared test was applied to compare observed and expected EdU indexes for each layer to determine the statistical significance of differences. For average number of FSCs, Fz3-RFP reporter intensity comparison, and EC/FSC ratio, a t-test was used to determine significance between indicated genotypes, and error was reported as standard error of the mean.

### Immediate FC method for calculating posterior FSC to FC conversion probability

The immediate FC method was used to calculate the probability for any layer 1 FSC to become an FC in a given cycle of egg chamber budding. As layer 1 FSCs directly give rise to FC daughters (*Reilein et al., 2017*), this was assessed by determining the proportion of germaria that contained a marked FC immediately posterior to the FSC region, which indicated recent FC production. This was only assessed in germaria with a small number of FSCs (1-3) to reasonably deduce the likelihood for an individual FSC. As the rate of proliferation would also influence this probability, this was accounted for in the immediate FC method equation.

Probability of a single layer 1 FSC becoming an FC in one cycle = p
Probability of a single layer 1 FSC dividing in one cycle = q

We assume that on average FSC division occurs halfway through a cycle, such that the probability of a newly-produced FSC becoming an FC is p/2. Therefore, the total probability (P) of an FSC becoming an FC in one cycle is the sum of two probabilities: an FSC becoming an FC and an FSC dividing and the additional FSC becoming an FC.

$p = p + (1-p)*q*(p/2)$
$p = p + pq/2 - p^2q/2$

We calculate the probability of an FSC becoming an FC by tallying the proportion of germaria (x) that do not have immediate FC daughters when a single marked layer 1 FSC is present (we assume that, on average, a single marked layer 1 FSC was present at the start of the prior cycle).

$$x = 1 – (p + pq/2 – p^2q/2)$$
$$x = 1 – p – pq/2 + p^2q/2$$
$$x = 1 – (1+q/2)*p + (q/2)*p^2$$
$$0 = (q/2)*p^2 – (1+q/2)*p + (1-x)$$
$$p = [(1+q/2) +/- SQRT((-(1+q/2)^2) – (4(q/2)(1-x)))]/q \text{ (using quadratic formula)}$$

Using this equation, we can solve for p, as x and q can be tabulated from scoring data.

We also considered germaria with immediate FC daughters that have no FSCs present in layer 1, as the only possibility is that a layer 1 FSC was present at the start of the last cycle and then became an FC. These instances were incorporated into the calculation of the proportion of germaria with a single layer 1 FSC but no immediate FCs.

If 2 or 3 FSCs were present, the square or cube root of the x ratio was used, respectively. A weighted average of adjusted x ratios for germaria with 1, 2 and 3 FSCs was calculated (weighted according to the number of examples of 0–1, 2, and 3 layer 1 FSCs) and used in the formula to calculate p.

The q value was adjusted based on measured proliferation for mutants compared to controls, as well as predicted daughter cell production, which assumes that seven FSCs (out of a total of 16 FSCs) are dividing per cycle to produce 5.6 FCs and 1.4 ECs per cycle. Therefore:

q = 'Proportion of EdU incorporation for layer 1 FSCs (mutant or control)' / 'average EdU incorporation of all control FSCs' * 7/16.

## EC per anterior FSC ratio

The EC/aFSC ratio was calculated by dividing the number of marked ECs per germarium present at 6d or 12d by the inferred average number of marked anterior FSCs per germarium during the 0-6d or 0-12d period. The inferred average number of anterior FSCs per germarium was the average of the observed number of anterior FSCs per germarium (at 6d or at 12d) and the observed number of anterior FSCs per germarium in the control samples at 6d for that experiment (representing our best estimate of the number of marked anterior FSCs per germarium at 0d in all samples of the same experiment). In all cases, only germaria that contained at least one marked FSC (in any position) were scored in order that there was an opportunity to produce new marked ECS throughout the experimental period. Results from multiple experiments for a particular genotype were aggregated by calculating the totals for marked ECs and for the inferred average number of anterior FSCs before deriving the overall EC per anterior FSC ratio.

## Acknowledgements

This work was supported by NIH RO1 GM079351 to DK. We thank Aaron Choi, Amy Reilein, Jamie Little, Jianhua Huang, Haoran Liu and Diana Kim for research assistance, Amy Reilein, Rachel Misner, Lena Kogan, Tulle Hazelrigg and Iva Greenwald for continued discussions and input, the Bloomington stock center for provision of genetic reagents, the Developmental Studies Hybridoma Bank (DSHB) for antibodies, FlyBase as an information resource, and the confocal microscope resource provided by the Department of Biological Sciences, Columbia University.

## Additional information

### Funding

| Funder | Grant reference number | Author |
| --- | --- | --- |
| National Institutes of Health | RO1 GM079351 | Daniel Kalderon |

The funders had no role in study design, data collection and interpretation, or the decision to submit the work for publication.

## Author contributions
David Melamed, Conceptualization, Resources, Data curation, Software, Formal analysis, Validation, Investigation, Visualization, Methodology, Writing - original draft, Writing - review and editing; Daniel Kalderon, Conceptualization, Resources, Data curation, Formal analysis, Supervision, Funding acquisition, Validation, Investigation, Visualization, Methodology, Writing - original draft, Project administration, Writing - review and editing

## Author ORCIDs
David Melamed (iD) https://orcid.org/0000-0002-2102-1340
Daniel Kalderon (iD) https://orcid.org/0000-0002-2149-0673

## Decision letter and Author response
Decision letter https://doi.org/10.7554/eLife.61204.sa1
Author response https://doi.org/10.7554/eLife.61204.sa2

# Additional files

## Supplementary files
• Transparent reporting form

## Data availability
All data analysed are included in the manuscript and supporting files. One source data file includes numerical data for all Figures.

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
