## [Decision Letter]

**Acceptance summary:**

This manuscript addresses the mechanisms by which stem cell domains can be spatially defined by extracellular signals. The authors show, using *Drosophila* Follicle Stem cells as a model system, that Wnt and Jak-Stat signaling controls positional patterns of cell division through signaling gradients. This work will be of interest to biologists in a number of fields including cell proliferation, cellular signaling, stem cell biology, and developmental biology.

**Decision letter after peer review:**

Thank you for submitting your article "Opposing Wnt and JAK-STAT signaling gradients define a stem cell domain by regulating differentiation at two borders" for consideration by *eLife*. Your article has been reviewed by two peer reviewers, and the evaluation has been overseen by a Reviewing Editor and Marianne Bronner as the Senior Editor. The following individuals involved in review of your submission have agreed to reveal their identity: Slobodan Beronja (Reviewer #1); Alana O'Reilly (Reviewer #2).

The reviewers have discussed the reviews with one another and the Reviewing Editor has drafted this decision to help you prepare a revised submission.

Summary:

This study delineates signaling gradients that regulate stem cell fate choice between proliferation (renewal) and differentiation using *Drosophila* follicular epithelium as a model. Building on published work that implicated anterior Wnt and posterior JAK/STAT as regulators of follicular stem cell (FSC) behaviors, and new understanding of the size and dynamics of FSC population, they revisit the model using detailed, functional and quantitative lineage tracing analyses. Overall ,this is a remarkably thorough study, employing a large number of KO and overexpression alleles, both singly and in parallel. The phenotypic analyses are quantitative in nature, were well done and conclusions drawn are well supported and thought through.

Essential revisions:

The reviewers' major concerns were the narrow focus of the text relative to the rest of the field. The reviewers felt that some of the conclusions were overstated and that the nomenclature obfuscated the significance of the findings, as well as a habit of self-citation that reduces impact due to lack of a context within the broader body of work done in this field. They agreed that these weaknesses can be addressed with writing changes. Specifically, the reviewers thought you should:

1) Describe the significant conceptual advancement of this study is (e.g. new players, principles, interactions, downstream mechanisms etc.).

2) Nomenclature: Despite clear demonstration in this work and the two prior papers from the same lab that fate outcomes, proliferation rates, and other characteristics differ between cells within the three layers of focus, they are all termed "FSCs". This terminology counters the data presented that Layer 3 cells produce escort cells as a primary function. Just because a cell CAN generate a follicle cell does not make it an FSC. In another example, the claim that cells in Layer 1 fated for differentiation are FSCs is not well supported. If the cells have completed division, and sit there waiting for the opportunity to interact with a germline cyst to leave the niche, it is not clear that they retain the capacity to self-renew, the defining characteristic of a stem cell. Finally, introduction of the term "Immediate FCs" creates a new category of cell without context to prior nomenclature. Are these cells the same as "pre-FCs" that have been referred to previously? It is not quite clear why Layer 1 cells that don't divide and are fated to differentiate into FCs are not "Immediate FCs" as well. This work really focuses on defining position effects on cell fate outcomes, with strong data supporting the concept that where a cell is located has a tremendous influence on its function and fate. Given the significant effort here to clarify the functional roles of cells in the three layers of focus, there is an opportunity to establish terminology that will accurately reflect cellular characteristics and function in the context of positioning, rather than introducing additional complication.

3) The dynamic nature of germline cyst development is not factored into the conclusions. Does altering JAK/STAT, Wnt, Hippo, or cell cycle signaling impact how and when the relevant somatic cells interact with germline cysts? Some of the data may depend on whether a germline cyst just passed through the FSC niche, dragging cells from one or more Layers through as part of the newly developing epithelium. This is particularly important in cases where the gradients were flattened or "rescued" by genetic manipulations that this same lab has previously demonstrated have a dramatic effect on adhesion molecules and interaction with germ cells. Similarly, the conclusion that the effects are autonomous may be overstated, particularly if the manipulations influence germ cell-somatic cell interactions.

4) Along the same lines, germaria bearing manipulation of these key signaling pathways exhibit dramatic morphological defects (Figures 2E, 3G both with increased Hop are particularly strong examples- (the germarium is facing the wrong way in 3G)). Conclusions are drawn with the expectation that combination genetic backgrounds (e.g. combining STAT and Hpo or Hop and Dacapo) simply alter proliferation without consideration of the global changes caused by these mutations on niche structure, cellular morphology, cell-cell interactions, and transcriptome-wide changes in gene expression, in addition to proliferation. Could some of the unexpected findings be explained by crosstalk between manipulated pathways such that increased signaling in one intersects molecularly with decreased signaling of another, resulting in the observed outcome independently of proliferation? As the large number of pathways previously identified by this lab as altered under similar conditions ALL affect proliferation in addition to other cellular functions, the "yes-no" conclusions from these experiments are overstated. Placement of the results in a more considered context would strengthen the manuscript.

5) The maintenance of production and elimination of ECs is interesting, but not explained. ECs were divided into posterior and anterior compartments for scoring, but the actual impact of genetic manipulation of Layer 3 cells was not presented separately for the two pools. Differential source cells, proliferation rates, germ cell interaction (mentioned in passing in the subsection “EC turnover rate is influenced by Wnt and JAK-STAT pathway activities”), or mobility of anterior ECs may help explain the maintenance of a 1:1 ratio of production and loss of the ECs. Alternatively, forms of cell death other than apoptosis might contribute. A vital stain with a nuclear marker such as propidium iodide or similar might uncover cell death independent of apoptosis.

6) The results describing the impact of proliferation on Layer 1 cell conversion to FCs is very confusing. Controls convert at a rate of 61.6%. Altered conversion by 10-13% in either direction upon manipulation of *CycE*, or *smo*, or *yki* is not considered a change, but overexpression of Hop yielding a 78% is emphasized and then followed up with the complex, and difficult to interpret *UAS-Hop*; *UAS-dacapo* experiment. Statistical analysis for most of the data in this section is not presented causing conclusions to be overstated. A similar challenge arises in Figure 4 (e.g. E) where statements like "a gradient is maintained" are made, but the control comparison for the statistical analysis is not indicated or no statistical significance is indicated.

7) The significance of the Fas3 expression is unclear. It can be ectopically expressed in ECs, but then it also seems to promote conversion of Layer 1 cells to follicle cells. How can the Layers and FSCs be identified in genetic situations where Fas3 extends far to the anterior?

8) Calling the *arr* mutant condition "null" is not well supported, given the 39% retention of reporter expression in that context. If the "background" of the reporter is 39% of WT, there are technical issues that must be addressed. As such, conclusions based on this assumption are overstated.

9) The paper is excessively long, with extensive redundancy between the Results and the Discussion.

10) Scientists in this field are developing a trend of only citing their own papers rather than including and acknowledging the full literature. This negative trend diminishes the work, devalues the work of others, and results in a failure to place the results presented in the context of the field, focusing instead only on the context of an individual lab. Appropriate referencing is essential for this manuscript before it will be acceptable for publication.

---

## [Author Response]

Essential revisions:The reviewers' major concerns were the narrow focus of the text relative to the rest of the field. The reviewers felt that some of the conclusions were overstated and that the nomenclature obfuscated the significance of the findings, as well as a habit of self-citation that reduces impact due to lack of a context within the broader body of work done in this field. They agreed that these weaknesses can be addressed with writing changes. Specifically, the reviewers thought you should:1) Describe the significant conceptual advancement of this study is (e.g. new players, principles, interactions, downstream mechanisms etc.).

The main advance is encapsulated in the title and summary, namely an understanding of how a stem cell domain can be defined spatially by extracellular signals. We have altered the summary and Introduction to make clear that this question is a widely applicable and largely unanswered question for stem cell communities maintained by population asymmetry (likely including the majority of types of stem cell and certainly very important mammalian gut and epidermal stem cells). Secondary significant advances concern the spatial organization of division rates by JAK-STAT signaling, the co-ordination of stem cell division rate and the frequency of the major differentiation process by a single pathway (JAK-STAT), evidence of concerted influences of Wnt and JAK-STAT pathways with different potencies according to location (Wnt anteriorly, JAK-STAT posteriorly), further evidence supporting our current concept of FSC organization and behavior, and elucidation of specific mutational responses that can lead to clinically relevant outcomes regarding cancer initiation or stem cell dysfunction. We have used this request as a cue to clarify and emphasize these major points in several locations, including summarizing these issues in the Discussion.

2) Nomenclature: Despite clear demonstration in this work and the two prior papers from the same lab that fate outcomes, proliferation rates, and other characteristics differ between cells within the three layers of focus, they are all termed "FSCs". This terminology counters the data presented that Layer 3 cells produce escort cells as a primary function. Just because a cell CAN generate a follicle cell does not make it an FSC. In another example, the claim that cells in Layer 1 fated for differentiation are FSCs is not well supported. If the cells have completed division, and sit there waiting for the opportunity to interact with a germline cyst to leave the niche, it is not clear that they retain the capacity to self-renew, the defining characteristic of a stem cell. Finally, introduction of the term "Immediate FCs" creates a new category of cell without context to prior nomenclature. Are these cells the same as "pre-FCs" that have been referred to previously? It is not quite clear why Layer 1 cells that don't divide and are fated to differentiate into FCs are not "Immediate FCs" as well. This work really focuses on defining position effects on cell fate outcomes, with strong data supporting the concept that where a cell is located has a tremendous influence on its function and fate. Given the significant effort here to clarify the functional roles of cells in the three layers of focus, there is an opportunity to establish terminology that will accurately reflect cellular characteristics and function in the context of positioning, rather than introducing additional complication.

We appreciate the necessity to explain more thoroughly the bases for our designation of FSCs and therefore took the following steps. We added to the Introduction a brief history of investigating FSCs, including the fundamental reason that two decades of dogma had presented an incorrect picture and the nature of the methods used to determine the locations and total number of FSCs in a germarium. We return to the issue of FSC locations in the last section of the Discussion to spell out how the results of the current study provide further support specifically for FSCs being located in layer 1. A clarification of the designation “immediate FC” has been added to the section of Results where this term is introduced. The key responses to the specific reviewer issues raised are also presented below.

The definition of terms and translation of definitions to lineage studies are the first crucial issues with regard to FSCs. FSCs are defined as upstream cells that produce FCs throughout the lifetime of a fly. “Upstream” is easily demonstrated experimentally because FCs are defined by continued association with germline cysts and therefore necessarily have a limited, known lifetime. In Reilein et al., 2017 FSC lineages were therefore defined experimentally by the inclusion of marked FCs at least 9d after genetic cell marking (greater than the maximum 5d lifetime of FCs, even taking into account possible biological variability among samples). These FSC lineages were generated in single colors or multiple color combinations in different experiments and examined by scoring the location of all marked cells in comprehensive archived z-sections with a key spatial marker (Fas3) of AP location in the germarium to deduce a number of different properties.

The location of FSCs was deduced by examining FSC lineages where only a single labeled cell was present in territory that could plausibly harbor an FSC. FSCs were found at all possible radial locations in contact with the germarial wall, consistent with the absence of any known radial asymmetry in the germarium. In about 50% of cases labeled cells were only in layer 1 (immediately anterior to Fas3), in almost 40% of cases labeled cells were only in layer 2 and in 10-15% of cases labeled cells were only in layer 3. Thus, cells in each of the three layers are FSCs, defined by their ability to produce FCs and being longer-lived than FCs.

Because FSCs were found at all radial positions and were shown by live imaging to undergo extensive radial movements it is likely that all cells in layer 1 (8, on average) and layer 2 (6, on average) are FSCs (at least 14 in total).

The total number of FSCs was also deduced by (i) measuring the proportion of all FCs produced by a single FSC and (ii) counting the number of differently colored lineages in a single ovariole using multicolor lineage analysis. In each case there were significant additional considerations because two lineages can share the same color and because FSC lineages are lost over time. Taking these considerations into account, the best estimate from each approach was 16 FSCs per germarium. That number is consistent with deductions from FSC locations of 8 layer 1, 6 layer 2 and 2 layer 3 FSCs on average.

No other study has defined FSC locations by these unbiased and comprehensive experimental approaches that are based on the fundamental functional definition of an FSC.

Further studies showed that the FSC population is heterogeneous with *instantaneously* different proclivities according to their exact AP location (where FCs derive directly from layer 1, while ECs derive directly from layers 2 or 3). Importantly, however, FSCs can move from one layer to another and therefore form a single community, appropriately named FSCs because they all share responsibility for maintaining the production of FCs (and ECs). Lineages derived from a single FSC frequently include both marked ECs and FCs. Thus, a cell in a given layer at one instant can, and often does, initiate a lineage that gives rise to ECs and FSCs. Indeed, if lineages with a marked EC are assayed for the location of candidate FSCs, those FSCs (in different samples) are found distributed among the three layers, just as for lineages defined by FC production. The experimental results, which reflect the average behavior of FSCs over several days and assayed over many samples, show that both EC-producing and FC-producing FSCs map to the same three layers. Cells in all three layers are therefore appropriately termed FSCs and it is understood that any one FSC may support future production of FCs and/or ECs. There is no conflict between the observations that FSCs in layer 1 only instantaneously have the opportunity to become FCs, while anterior FSCs can only instantaneously become ECs, and the longer-term outcomes, which turn out to be very similar for FSCs in different locations due to the various stochastic options at play (division, differentiation and changing AP location). Hence, our results do not show that EC production is the primary function of layer 3 (and 2) FSCs- that is simply the only immediate differentiation option for anterior FSCs. Likewise, we provided clear evidence that layer 1 cells are FSCs but we never make any claim regarding “layer 1 cells fated for differentiation”; that idea contradicts our findings about stochastic behaviors. Because the behavior of each FSC is stochastic the range of behaviors of individual FSCs from a given instant in time onwards can also include becoming an FC without ever dividing or becoming an EC without ever producing an FC (and such cells are distinguished from FCs because they survive longer). These stochastic behaviors are characteristic of all stem cell communities maintained by population asymmetry. Our experimental observations in Reilein et al., 2017, did look at FSC-initiated lineages where there was at least one FSC and FC derivative 9 days later (FSCs that showed direct evidence of self-renewal) in order to determine FSC locations, so the location of FSCs described earlier was deduced from sampling self-renewing stem cells.

In summary, the term FSC describes the demonstrated functional properties of cells and the terms layer 1, 2 and 3 specify their exact AP locations within the stem cell domain.

The term “immediate FC” is not intended to define a specific category of FC. It is intended only to give spatial information. The germarium typically contains two germline cysts, with a stage 3 cyst entirely surrounded by Fas3+ cells and a stage 2b cyst with Fas3+ cells only around its posterior face. Which of the Fas3+ cells between the stage 2b and 3 cysts are associated with each cyst is hard to determine. As a surrogate for this distinction, we call the cells closest to the stage 2b cyst “immediate FCs”, indicating that they are the most recently produced. The remaining Fas3+ cells are scored as being in the germarium but not “immediate FCs”.

We have defined the terms we use and used them consistently. It is not possible to reconcile our terms with those used by some other groups because those studies either precede or do not acknowledge our evidence published in 2017 of a number and arrangement of FSCs very different from prior conceptions. Most FCs continue to divide until stage 6 and acquire differentiated characteristics at around stage 9, so the term “pre-FC” has sometimes been used to refer to earlier proliferative or undifferentiated states. We do not use that term and simply refer to all cells permanently associated with a germline cyst as FCs.

3) The dynamic nature of germline cyst development is not factored into the conclusions. Does altering JAK/STAT, Wnt, Hippo, or cell cycle signaling impact how and when the relevant somatic cells interact with germline cysts? Some of the data may depend on whether a germline cyst just passed through the FSC niche, dragging cells from one or more Layers through as part of the newly developing epithelium. This is particularly important in cases where the gradients were flattened or "rescued" by genetic manipulations that this same lab has previously demonstrated have a dramatic effect on adhesion molecules and interaction with germ cells. Similarly, the conclusion that the effects are autonomous may be overstated, particularly if the manipulations influence germ cell-somatic cell interactions.

Our data are collected from fixed images but they presumably sample evenly from different stages of germline cyst passage through the FSC region. The results therefore represent an average of all times during a roughly 12h budding cycle. An important part of our analysis is to determine how signaling pathways affect interactions with germline cysts by deducing the frequency with which layer 1 FSCs become FCs and we report those results comprehensively. We do not have further insights into exactly how changes in signaling pathways alter differentiation to FCs (other than preliminary evidence for participation of Fas3 under some circumstances).

As explained in the first section of Results to introduce the issue of measuring cell autonomous responses, clonal analyses mostly report on germaria with only a minority of FSCs, ECs or FCs labeled and harboring an altered genotype. In these cases it is unlikely that the behavior of other cells, neighboring unmarked somatic cells or germline cysts, are altered in response to differences in the labeled cells. There are some exceptions by 12d for genotypes that cause excessive accumulation of FSCs or ECs and in those cases there may be secondary consequences. However, the trends for measured behaviors at 12d in these cases are already apparent (and the same) at 6d with regard to EC production, FSC AP location and FSC numbers, while proliferation and conversion of layer 1 FSCs to FCs was always measured only at 6d.

For experiments with genetic manipulation of all FSCs and ECs using C587-GAL4 we deliberately used a short period of time (3d at the restrictive temperature that allows transgene expression) so that secondary effects would be minimized and the only quantitative data extracted were EdU incorporation frequencies, which are likely to be cell autonomous responses. Thus, almost all measurements reported are indeed cell autonomous (the properties of only genetically labeled cells were measured) and are unlikely to include significant contributions from secondary relays involving neighboring cells. The cell autonomous description is used not only to relay this methodological reality and appropriate interpretation of results but also to make clear that another genetic approach is possible in which all cells are simultaneously subjected to the same genetic manipulations over long time periods. That type of experiment is important and would likely involve many secondary and tertiary compensatory interactions but has not been conducted here.

4) Along the same lines, germaria bearing manipulation of these key signaling pathways exhibit dramatic morphological defects (Figures 2E, 3G both with increased Hop are particularly strong examples- (the germarium is facing the wrong way in 3G)). Conclusions are drawn with the expectation that combination genetic backgrounds (e.g. combining STAT and Hpo or Hop and Dacapo) simply alter proliferation without consideration of the global changes caused by these mutations on niche structure, cellular morphology, cell-cell interactions, and transcriptome-wide changes in gene expression, in addition to proliferation. Could some of the unexpected findings be explained by crosstalk between manipulated pathways such that increased signaling in one intersects molecularly with decreased signaling of another, resulting in the observed outcome independently of proliferation? As the large number of pathways previously identified by this lab as altered under similar conditions ALL affect proliferation in addition to other cellular functions, the "yes-no" conclusions from these experiments are overstated. Placement of the results in a more considered context would strengthen the manuscript.

The images referenced are the exceptions discussed above where overall morphology is altered. As described above, the properties of the exceptional genotype scored in MARCM experiments at 12d were also evident at 6d before the labeled cells had accumulated extensively. The germaria with altered morphologies referenced for experiments using C587-GAL4 were used only to score EdU frequencies (Figure 2E) or show ectopic Fas3 expression and ectopic encapsulated cysts qualitatively at later times (Figure 3G).

One purpose of testing *UAS-Hop* with *UAS-Dap* was to measure behaviors in the absence of complicating phenotypes due to a build-up of labeled cells by 12d. However, the major purpose of this test and the four different *stat* genotypes (with *kibra*, *wts*, *UAS CycE* or *kibra* and *UAS-CycE*) was to examine if changes seen in each behavioral parameter for *UAS-Hop* or *stat* were also observed when FSC proliferation was brought to a more normal range. The additional genetic changes were used as a device expected primarily to affect cell cycling (based on earlier studies). Each might *a priori* also affect other properties. We did assay FSC lineages with those genetic changes alone and show comprehensive results for *UAS-CycE* and the most pertinent results (EdU incorporation) for all; other measured properties, not shown, did not reveal significant differences from normal behavior, though it would be premature to rule out any influences of the Yki pathway. The results of the assays of compound genotypes were in line with results from *stat* or *UAS-Hop* alone with regard to behaviors other than proliferation. In several cases the consequences were more clear-cut because FSC division rates and FSC numbers were closer to normal. Thus, while the cell cycle and Yki pathway regulators used might have some pleiotropic effects our assays of those genetic alterations alone or in combination with changes in JAK-STAT signaling did not give any indication of effects beyond proliferation.

5) The maintenance of production and elimination of ECs is interesting, but not explained. ECs were divided into posterior and anterior compartments for scoring, but the actual impact of genetic manipulation of Layer 3 cells was not presented separately for the two pools. Differential source cells, proliferation rates, germ cell interaction (mentioned in passing in the subsection “EC turnover rate is influenced by Wnt and JAK-STAT pathway activities”), or mobility of anterior ECs may help explain the maintenance of a 1:1 ratio of production and loss of the ECs. Alternatively, forms of cell death other than apoptosis might contribute. A vital stain with a nuclear marker such as propidium iodide or similar might uncover cell death independent of apoptosis.

We have re-written these sections extensively and refined our methods for calculating and presenting EC production over 0-6d and 0-12d periods. The results clearly illustrate significant turnover of wild-type ECs produced from FSCs. We now express the numerical results in a way that makes clear that the high turnover cannot apply to the whole EC population, but only to ECs produced from FSCs during adulthood. Since submission we accumulated more results for lineages expressing *UAS-DIAP1* to inhibit apoptosis and are now able to report reliable results showing that apoptosis makes only a small contribution to turnover. There may indeed be other forms of EC cell death and that may be an important avenue for future investigation, although it is hard to measure rates of cell death quantitatively. From our results to date we suggest it is likely that turnover is due in part to cells moving from EC regions back to the FSC domain and we acknowledge that this remains to be demonstrated definitively. We also added more data for *arr* mutant cells expressing *UAS-DIAP1* to make reliable conclusions about the contribution of apoptosis to the greatly reduced accumulation of marked ECs lacking Wnt pathway activity.

We did indeed score region 1 and region 2a ECs separately for all samples but we did not find significant changes over time or genotypes that we thought would allow any conclusions, so we did not present EC production data separated into two categories.

6) The results describing the impact of proliferation on Layer 1 cell conversion to FCs is very confusing. Controls convert at a rate of 61.6%. Altered conversion by 10-13% in either direction upon manipulation of CycE, or smo, or yki is not considered a change, but overexpression of Hop yielding a 78% is emphasized and then followed up with the complex, and difficult to interpret UAS-Hop; UAS-dacapo experiment. Statistical analysis for most of the data in this section is not presented causing conclusions to be overstated. A similar challenge arises in Figure 4 (e.g. E) where statements like "a gradient is maintained" are made, but the control comparison for the statistical analysis is not indicated or no statistical significance is indicated.

We have removed the data for *cycE*, *smo*, *yki* because they do not provide a clear resolution of whether division rate affects conversion of FSCs to FCs. Loss of JAK-STAT activity was tested alone and using four additional genotypes with a variety of proliferation rates, including near-normal rates. In all cases, conversion of FSCs to FCs was greatly and significantly lower in the absence of STAT activity, as indicated in Figure 3C. Moreover, the additional expression of Fas3 in one genotype significantly increased FC production (p value also indicated in Figure 3C). Increasing JAK-STAT pathway activity with *UAS-Hop* significantly increased FC production frequency from layer 1 FSCs, while *UAS-Hop* plus *UAS-Dap* produced a similar increase that was not statistically significant (fewer experiments with that genotype). These statements of significance are now included in the text in addition to being presented in the figures.

In sum the *UAS-Hop*; *UAS-dacapo* experiments were designed for excess Dacapo to suppress the increased FSC division due to excess Hop. The results report that EdU indices were roughly normal for *UAS-HopUAS-Dap* FSCs, allowing measurement of various parameters (FSC AP location, FC production and FSC numbers) in response to increased JAK-STAT pathway, while FSC division rates were maintained at roughly normal levels.

The statistical significance of the reported changes in FSC to FC conversion for changes in Wnt signaling alone or compound changes are indicated in Figure 4B and Figure 6A and explained in the legend and Materials and methods.

Similarly, for all figures reporting FSC AP locations (Figures 3A, 4A, 6D) statistical significance of alterations in FSC locations are indicated and the explanation is now expanded in the legend; the proportion of FSCs in a specific layer is in each case compared to control values.

All figures reporting the pattern of EdU incorporation (Figures 2A, 4E, 5) included designations of statistical differences from controls for each layer. The exception is in Figure 5A where the effect of *dnTCF* is specifically isolated. Additionally, in Figure 2K we present statistical significance of differences between neighboring layers for all *stat* mutant genotypes as support for the claim that the pattern of EdU indexes among layers remain graded.

We now performed an additional set of calculations (with p values explicitly in source data files and the method added to the statistics section) for all genotypes that determines the statistical significance of differences from controls in the EdU index of each layer relative to the whole FSC population for that genotype. Those calculations are presented by # symbols or figure legend explanations of no significant differences. Specifically, significant differences were found for *UAS-Hop* in Figure 2A, *UAS-Hop* and *UAS-CycE* in Figure 2J. No significant differences were found for *stat* FSCs in Figure 2A, for any genotype in Figure 2K, Figure 4E or Figure 5A, providing support for the statements that there patterns of EdU incorporation were not significantly different from controls for all genotypes lacking STAT activity and all genotypes with reduced or absent Wnt pathway activity.

7) The significance of the Fas3 expression is unclear. It can be ectopically expressed in ECs, but then it also seems to promote conversion of Layer 1 cells to follicle cells. How can the Layers and FSCs be identified in genetic situations where Fas3 extends far to the anterior?

In experiments where ectopic Fas3 is induced by expressing *UAS-Hop* conditionally with C587-GAL4 by shifting to 29C, FSC locations were scored after 3d when measuring the EdU index. At that time there is only sporadic or no ectopic Fas3 and it is therefore easy to recognize the normal anterior border of strong Fas3 expression. Only later, as in Figure 3G, is ectopic Fas3 too extensive to determine the normal Fas3 border. In the majority of germaria of MARCM clones expressing *UAS-Hop* there are large numbers of unlabeled ECs, FSCs and FCs, allowing the normal landmarks of the germarium to be recognized. There were some germaria where the number of labeled cells is so high, as in Figure 3F that the normal Fas3 border cannot be recognized definitively. In these rare cases, a combination of Fas3 pattern, germline cyst locations and overall morphology were used to make a best estimate of FSC locations.

In experiments where UAS-Fas3 was expressed in MARCM clones deficient for STAT the normal anterior Fas3 border was easily recognized because only a small number of labeled cells were present and those cells showed similar Fas3 expression to their lateral neighbors. Even though excess JAK-STAT produces ectopic Fas3 there is still Fas3 expression in *stat* mutant FCs and Fas3 is not readily detected in FSC locations for *stat* UAS-Fas3 cells. Hence, there was no ambiguity in scoring and we report a genetic result of the effects of UAS-Fas3 in a STAT-deficient setting. To acknowledge that there is much more to explore with regard to the regulation and role of Fas3 expression we added a sentence to that effect.

8) Calling the arr mutant condition "null" is not well supported, given the 39% retention of reporter expression in that context. If the "background" of the reporter is 39% of WT, there are technical issues that must be addressed. As such, conclusions based on this assumption are overstated.

The re-phrasing of these sentences clarifies that the *arr* allele used is considered a null and that Arr is absolutely required for canonical Wnt signaling based on prior investigations (with the key reference added), and that the residual RFP signal plausibly results from some combination of perdurance of RFP protein and RNA as functional Arr product is being diluted out in the clone, together with Fz3-RFP expression that is not dependent on Wnt pathway activity. The reporter has previously been shown to respond to changes in Wnt pathway activity but there is no reason to believe that it has zero expression in this or any other tissue in the absence of Wnt signaling. We added analogous studies for a STAT-RFP reporter since submission of the first manuscript and found analogous results- significant residual RFP levels in MARCM clones that used a *stat* null mutation. The prior genetic evidence regarding all of these alleles provides a sound basis for our phrasing of eliminating pathway activities and the phenotypes we describe are consistent with that (extreme phenotypes for *stat* and stronger phenotypes for *arr* than for *dnTCF*). Residual reporter activity is higher than we might have expected but there are plausible explanations for those findings.

9) The paper is excessively long, with extensive redundancy between the Results and the Discussion.

We have edited extensively, including deletion of large segments repeating the evidence for the major findings concerning patterned proliferation and differentiation changes in response to the two pathways, as well as evidence relating to EC turnover. The Discussion now repeats central findings in summary form and discusses new issues, including more discussion of the relevance of our results to other stem cell paradigms and further explanation of the status of different portrayals of FSC organization, as requested in other comments.

10) Scientists in this field are developing a trend of only citing their own papers rather than including and acknowledging the full literature. This negative trend diminishes the work, devalues the work of others, and results in a failure to place the results presented in the context of the field, focusing instead only on the context of an individual lab. Appropriate referencing is essential for this manuscript before it will be acceptable for publication.

We now discuss the relevance of our work in a wider context of adult stem cells, with special reference to those stem cells organized similarly to FSCs, specifically featuring mammalian intestinal stem cells. We have added numerous references to earlier work regarding signaling pathways that influence FSCs and observations relevant to ECs. A key attribute of our study is that we are assaying multiple FSC behaviors in a quantitative manner and using the insights gained from comprehensive studies of wild-type FSC behavior that drastically altered conception of FSC numbers, locations and activities. No studies from other labs have assayed these individual behaviors in comparable detail; papers published prior to 2017 could not possibly incorporate our revised picture of FSCs and assays of these individual FSC behaviors, while several studies published more recently have chosen not to do so. We cannot therefore usefully discuss most previous findings concerning FSCs with an appropriate level of detail because only crude measures of FSC survival were generally made and only a subset of FSCs were identified and assayed. Accordingly, we have instead discussed prior FSC models and the evidence that distinguishes these from our conception. We also now describe how specific results obtained in this study provide further support for our current picture of FSC organization and how some of the outcomes we observed may be of general relevance to diseases originating from stem cell mutations.